# Elasto-magnetic instabilities for amplified actuation and mechanical memory

Seong-Yu Choi [1,7], Ji-Sung Park [2,3,7], Won Jun Song[1], Maga Kim[1], Yun Hyeok Lee [1], Yong Eun Cho[1], Hakjun Lee[1], Ho-Young Kim [2,4] & Jeong-Yun Sun [1,5,6]

Instabilities can generate fast and amplified motion in mechanical systems. Here, we present an elasto-magnetic instability that combines magnetic attraction and elastic tension to create bistable dynamics. To demonstrate this, we built a coupled elasto-magnetic vibration system that produces amplified motion and greater displacement and force than a control system across a wide frequency range. We also establish design principles that can be applied to different configurations by studying the balance between magnetic and elastic forces. The system also shows inertial hysteresis, which enables mechanical memory by storing external inputs in volatile and non-volatile modes with adjustable thresholds. This dual function of amplification and memory shows how instabilities can be potentially used for programmable and adaptive soft actuation.

Mechanical systems are often constrained by their intrinsic material properties[1]. In soft materials, this manifests as low mechanical strength and slow response times[2], posing challenges for high-performance actuation. Yet, nature offers numerous examples—such as the Venus flytrap, bladderwort, and pistol shrimp—that overcome these limitations through instability-driven motion, in which elastic energy is stored and rapidly released to produce fast and powerful movements[3–7].

Inspired by such strategies, soft actuators have been engineered to incorporate structural instabilities (e.g., buckling, wrinkling, snap-through) through the use of intrinsically soft materials or compliant architectures[8–18]. These designs allow such systems to generate forces far exceeding their passive material limits. Recent developments have extended such instability-based actuation to diverse applications, including sensors[19], logic devices[20,21], and metamaterials[22–24], activated through pneumatic[12,25,26], electromagnetic[24,27,28], electrical[29,30], and chemical[13,31] stimuli.

Instability-based mechanisms inherently involve rapid, large transitions between distinct states. During these transitions, inertia—the tendency of a body to maintain its motion—is a central element of the dynamics. In structural dynamics[32,33] and in the broader literature on nonlinear phenomena[34,35], inertia governs transient responses and subsequent resettling. In soft actuation, however, it has often been treated as a passive consequence rather than an explicit design parameter. Although prior studies have shown that inertia can amplify motion[36,37], its deliberate use to drive transitions between coexisting states or to sustain motion is comparatively less explored. Harnessing both inertia and mechanical instabilities, therefore, offers a practical design strategy for soft actuators with improved adaptability and energy efficiency.

Here, we introduce an Elasto-Magnetic Instability (EsMI) that couples inertia-driven motion with tunable bistability, achieved through the interplay between magnetic attraction and elastic tension. To realize this mechanism, we developed a Coupled Elasto-Magnetic Vibration (C-EsMV) system that integrates permanent magnets and an elastic membrane, forming a bistable architecture capable of enhancing kinetic energy conversion by over three orders of magnitude compared to a non-coupled control system (NC-EsMV) (Fig. 1a, b). This design quantifies and leverages the balance between characteristic magnetic

[1]Department of Materials Science and Engineering, Seoul National University, Seoul, Republic of Korea. [2]Department of Mechanical Engineering, Seoul National University, Seoul, Republic of Korea. [3]Institute of Mechanical Engineering, École Polytechnique Fédérale de Lausanne, Lausanne, Switzerland. [4]Institute of Advanced Machines and Design, Seoul National University, Seoul, Republic of Korea. [5]Research Institute of Advanced Materials (RIAM), Seoul National University, Seoul, Republic of Korea. [6]SNU Energy Initiative, Seoul National University, Seoul, Republic of Korea. [7]These authors contributed equally: Seong-Yu Choi, Ji-Sung Park. ✉e-mail: hyk@snu.ac.kr; jysun@snu.ac.kr

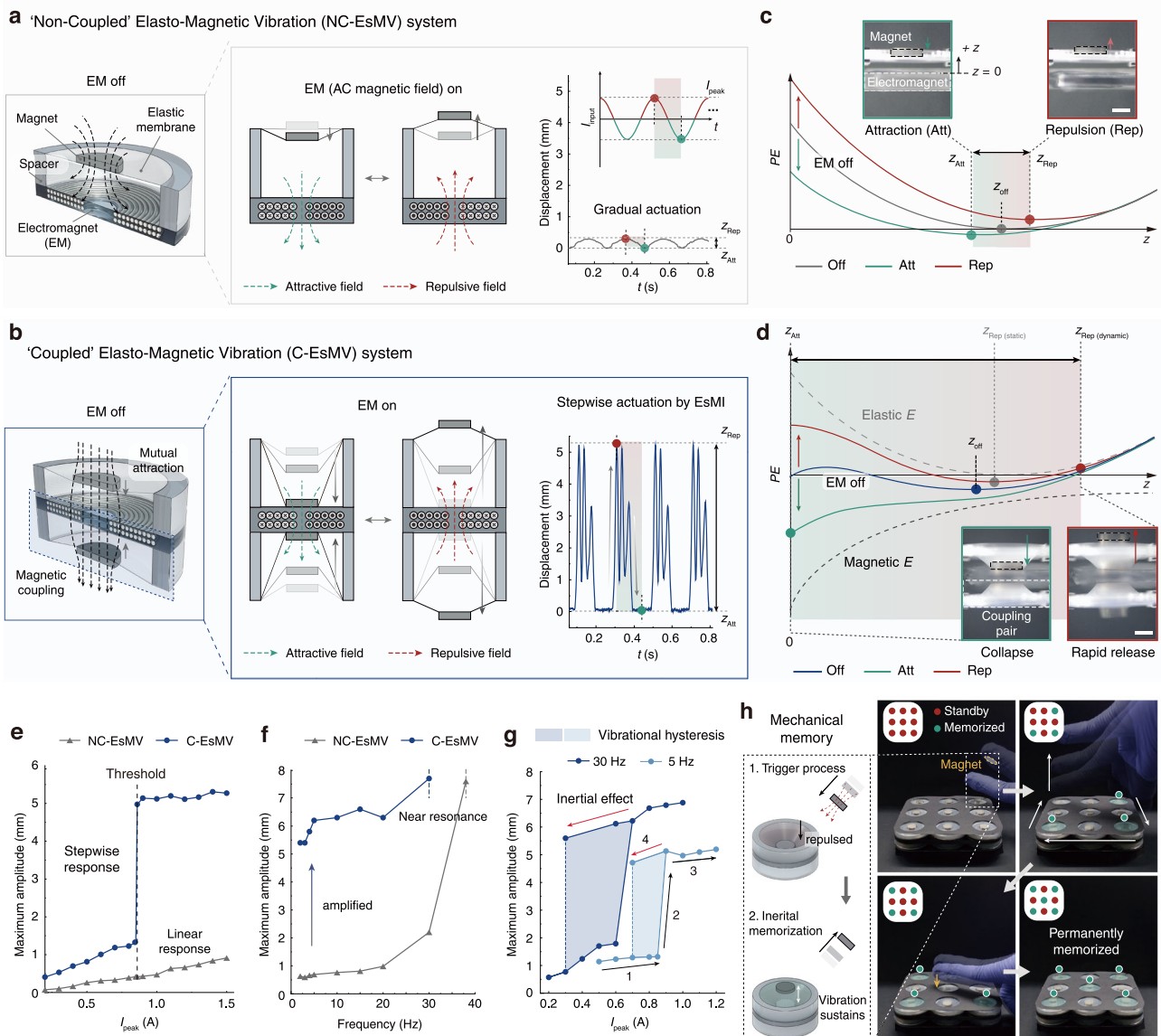

**Fig. 1 | Coupled elasto-magnetic vibration (C-EsMV) system enabling amplified actuation and mechanical memory. a, b** Configurations of elasto-magnetic vibration (EsMV) systems: **a** non-coupled (NC-EsMV) and **b** coupled (C-EsMV). When an AC is applied to an electromagnet, permanent magnets vibrate due to cycles of attractive (green dashed arrow) and repulsive (red dashed arrow) field. Even with the same input current, the two systems exhibit entirely different vibration behaviors ($f_i = 5$ Hz, $I_{peak} = 0.86$ A, Scale bar: 3 mm). **c, d** Schematic representations of potential energy landscapes for **c** NC-EsMV and **d** C-EsMV. Under alternating magnetic fields (attractive/ repulsive) applied by an electromagnet, the potential energy oscillates from each system's equilibrium state ($z_{off}$), driving vibrations. In C-EsMV, the elastic potential energy of the membrane (gray dashed line) and the magnetic attraction between coupled magnets (black dashed line) combine to create Elasto-magnetic instability. The large inertia in C-EsMV allows the magnet to overshoot the static equilibrium position, extending from $z_{Rep, (static)}$ to $z_{Rep, (dynamic)}$, resulting in vibration amplitudes significantly larger than NC-EsMV. **e–g** Key features of the C-EsMV system: **e** Maximum vibration amplitude as a function of input current, showing a nonlinear, stepwise response in C-EsMV compared to the linear response in NC-EsMV ($f_i = 5$ Hz). **f** Vibration amplitude across frequencies, with C-EsMV maintaining large amplitudes from low frequencies to near resonance ($I_{peak} = 1$ A). **g** Vibrational hysteresis by inertia, where amplified vibration persists below the current threshold. This effect is more pronounced near resonance ($f_i = 30$ Hz) due to increased energy absorption. **h** Non-contact, non-volatile mechanical memory arrays. When a magnetic trigger is applied to each cell, it transitions from the *standby* to the *memorized* state, permanently recording the external mechanical trigger ($f_i = 30$ Hz, Scale bar: 2 cm).

($F^*_{MM}$) and elastic ($F^*_{ES}$) forces, establishing a scalable framework for inertia-enhanced actuation. Incorporating an electromagnet further enables real-time modulation of the instability landscape, leading to reconfigurable transitions and nonlinear, stepwise amplification with vibration-induced hysteresis. This hysteretic behavior enables mechanical memory, allowing external triggers to be encoded into volatile and non-volatile modes with tunable activation thresholds.

## Results

### Elasto-magnetic instability (EsMI)

When two magnets with identical polar orientations approach each other, their attractive force rises sharply, leading to the collapse-like motion of magnets (Supplementary Fig. 1a). They accelerate toward each other and adhere, settling into the lowest-energy configuration. Although this magnetic coupling has the potential to utilize the inherent strength of magnets, due to its accelerating nature when attracted, it has been mainly used in applications such as array[38], field alignment[39], and structure reconfiguration[40,41]. To harness the powerful magnetic

motion, we adopt an elastic membrane acting as a spring, capable of storing and releasing energy (Supplementary Fig. 1b). The resulting combination of magnetic and elastic force generates an elasto-magnetic instability (EsMI) with two stable states (Fig. 1d, blue line).

## Elasto-magnetic vibration (EsMV) systems

To validate the instability arising from magnetic collapse, two types of elasto-magnetic vibration systems were developed: one consists of a single elastic membrane-magnet composite positioned above one side of the electromagnet, with acrylic spacers to adjust the gap (NC-EsMV, Fig. 1a). The other employs two membrane-magnet composites on either side of the electromagnet, coupled through mutual attraction (C-EsMV, Fig. 1b). The electromagnet, driven by a sinusoidal input alternating current (AC) signal, generates a magnetic field that cyclically attracts and repels the magnet(s), inducing oscillatory motion (Supplementary Fig. 2). This motion stretches the membrane, producing an elastic counterforce.

To analyze the basic behaviors of both systems, we calculated the elasto-magnetic potential energy and the corresponding equilibrium positions as a function of input current. As shown in Supplementary Fig. 3, the NC-EsMV system remains monostable over the entire current range (−2.0 A to +2.0 A), and the equilibrium position changes only slightly with current ($\Delta z/D \approx 0.25$). In contrast, the C-EsMV system exhibits bifurcation in behavior, so that the energy landscape alternates between monostable and bistable regimes as the current varies, leading to a much larger shift in equilibrium position ($\Delta z/D > 1$). The appearance of a bistable regime indicates the presence of an elasto-magnetic instability (EsMI), arising from the magnetic attraction between the paired magnets and the elastic restoring force of the membranes. This distinction is directly reflected in the vibration behavior. In NC-EsMV, the magnets' natural collapse motion is not utilized; the system relies solely on the electromagnet to drive magnet's movement, resulting in only small-amplitude vibrations between $z_{Att}$ and $z_{Rep}$ (Fig. 1a). This behavior is consistent with the potential energy graph, where the magnet's equilibrium position, determined by the minimum of the potential energy, changes only slightly along the $z$-axis during each cycle of electromagnetic attraction and repulsion (Fig. 1c).

In C-EsMV, however, the system inherently exhibits a strong magnetic attraction between the magnet-membrane composites even without an applied electromagnetic field (i.e., EM field off, $z = z_{off}$), resulting in an energy barrier within the potential energy landscape. When an attractive electromagnetic field is applied, the magnets are drawn closer until reaching a critical distance, at which point they collapse onto the electromagnet surface, storing potential energy in the membrane ($z = z_{Att}$). Upon switching to a repulsive field, the equilibrium shifts from $z_{Att}$ to $z_{Rep, (static)}$, where the stored energy is rapidly converted into kinetic energy, yielding significantly larger vibrations compared to NC-EsMV (Fig. 1b and d and Supplementary Video 1). In addition, the composite's initially high inertia further amplifies the vibration amplitude, extending the peak position from $z_{Rep, (static)}$ to a dynamically overshoot position $z_{Rep, (dynamic)}$.

A noteworthy characteristic of C-EsMV is its nonlinear, stepwise response to input signals (Fig. 1e). For example, when $I_{peak} \leq 0.85$ A, both systems exhibit small-amplitude vibrations. However, a slight increase to 0.86 A causes C-EsMV to transition to high amplitude vibrations, while NC-EsMV remains nearly unchanged. Achieving large outputs with small input changes is crucial for system efficiency, and this threshold value can be readily tuned through structural design adjustments.

## Resonance effect and hysteric behavior

Elasto-magnetic vibrations, driven by the electromagnet's signal, are governed by both the membrane's natural frequency and the external signal frequency. In NC-EsMV, vibrations intensify near resonance due to increased energy absorption but remain confined to a narrow frequency range (Fig. 1f). In contrast, C-EsMV—enabled by the EsMI—sustains large-amplitude vibrations over a broader frequency range, including low frequencies in particular, thereby enhancing versatility. At resonance, the input energy is accumulated coherently with the oscillation, whereas beyond resonance, phase mismatch suppresses energy coupling, leading to the eventual decay observed experimentally (detailed in Supplementary Note 3).

Another distinctive feature of C-EsMV is its vibrational hysteresis (Fig. 1g and Supplementary Video 2). Once triggered, the system retains amplified motion even as the input current falls below the critical threshold for amplification, exhibiting behavior analogous to a Schmitt trigger[25] under both low- and relatively high-frequency conditions ($f_i = 5$ Hz and 30 Hz). This retention originates from the inertia of the magnets, which sustains continuous exchange between elastic and kinetic energy once amplification is initiated. The system maintains stable vibration amplitudes over 60,000 cycles (Supplementary Fig. 4), demonstrating its ability to preserve performance under reduced energy input. Near resonance (~30 Hz), this hysteretic effect becomes more pronounced, highlighting the roles of acceleration and inertia in maintaining dynamic stability.

This inertia-driven mechanism allows us to build a mechanical memory, which "memorizes" external triggers as amplified vibrations without additional electrical input. As illustrated in Fig. 1h, a magnet attached to a finger repels the magnet on the membrane, leading to the storage of additional elastic potential energy. Once the finger is removed, the system keeps vibrating. This non-contact magnetic trigger shifts the system from a *standby* state (weakened vibration) to a *memorized* state (amplified vibration). The *memorized* state remains stable indefinitely unless the electromagnet is off or the system is intentionally reset (Supplementary Video 3). Further details of this behavior are discussed in Fig. 5, focusing on the application of mechanical memory.

## Mechanism of EsMV systems

To elucidate the magnetic coupling effects in the EsMV system, we analyzed the forces and energy exchanges during actuation using an analytical model detailed in Supplementary Notes 1 and 2.

We tracked the time evolution of magnets' position to capture acceleration and the consequent inertial effects. In Fig. 2, potential energy (blue line) and kinetic energy (red line) were plotted over time, with the magnet's position marked by yellow circles at characteristic moments. In the NC-EsMV system (Fig. 2a, b), the magnet's position ($z$) is measured from the electromagnet surface and the equilibrium position at the off-state corresponds to the initial position $D$ ($z_{off} = D$). During the repulsive phase (Fig. 2a), total potential energy increases ($t_0 \leq t \leq t_2$), displacing the magnet in the positive z-direction with minimal kinetic energy (~$10^{-4}$ mJ). During the attractive phase (Fig. 2b), the potential energy decreases ($t_2 \leq t \leq t_4$), pulling the magnet back with a negligible overshoot (~0.2 mm) owing to low inertia, as indicated by the shaded area.

In contrast, in the C-EsMV system (Fig. 2c, d), the mutual attraction between the magnets draws them slightly closer at the off-state ($z_{off} < D$). When magnetic and elastic energies are appropriately balanced, they create an energy barrier between the upper ($z = z_{off}$) and lower equilibrium ($z = 0$) points. This barrier allows energy accumulation until a critical threshold is reached, resulting in time-varying "snap-through" behavior. During the repulsive phase ($t'_0 \leq t < t'_2$), the magnets remain at the bottom surface (Fig. 2c) until the electromagnetic field dissipates the barrier. Once the barrier is removed, the stored elastic energy accelerates the magnet ($t'_2 \leq t \leq t'_3$), causing it to overshoot the upper static equilibrium by ~3 mm due to significant inertia. During the attractive phase ($t'_4 \leq t \leq t'_6$), the magnet returns to $z = 0$, completing the cycle (Fig. 2d).

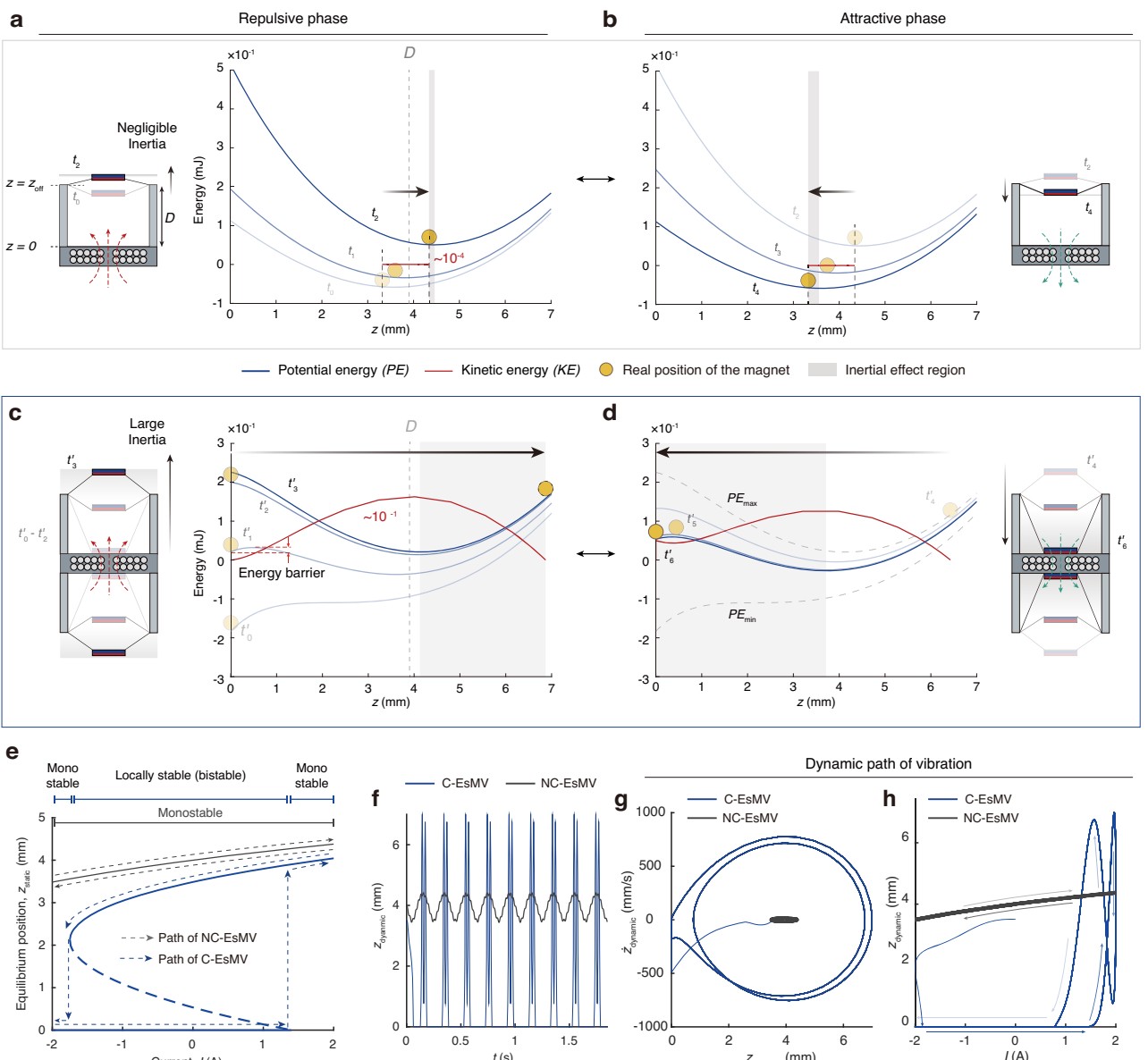

**Fig. 2 | Mechanism of an amplification in EsMV systems. a, b** Energy diagrams of NC-EsMV during the **a** repulsive phase ($t_0$–$t_2$) and **b** attractive phase ($t_2$–$t_4$). Kinetic energy (red) and potential energy (blue) are plotted over time. The yellow ball indicates the magnet's actual position, considering its inertia. In NC-EsMV, the magnet oscillates around static equilibrium points ($z = z_{off}$) with low kinetic energy ($\sim 10^{-4}$ mJ). **c, d** Energy diagrams of C-EsMV during the **c** repulsive phase ($t'_0$–$t'_3$) and **d** attractive phase ($t'_4$–$t'_6$). When this barrier is overcome by the combined effects of elastic restoration force and electromagnetic repulsion, the magnet is launched

with significantly higher kinetic energy ($\sim 10^{-1}$ mJ), resembling a slingshot motion ($t'_2$–$t'_3$). **e–h** Static and dynamic vibration analyses of NC-EsMV and C-EsMV systems. **e** Motion trajectories of each system derived from bifurcation diagram analysis. **f** Time-dependent vibration amplitudes showing dynamic responses of each system. **g** Phase portraits illustrating distinct dynamic states. **h** Dynamic vibration trajectories under sinusoidal input. All analysis was conducted representatively at $f_i = 5$ Hz and $I_{peak} = 2.0$ A.

Throughout a cycle, NC-EsMV exhibit small oscillations with net force and kinetic energy remaining low. The low net force and kinetic energy stem from inefficient energy transfer, as most of the electromagnet-generated potential energy dissipates into the surrounding air rather than converting effectively into kinetic energy. Consequently, the NC-EsMV system results in limited energy exchange and subdued motion. In contrast, the C-EsMV system establishes a critical threshold that enables efficient conversion of electrical input into mechanical motion. This configuration enables improved conversion efficiency through elasto–magnetic coupling, allowing a larger portion of the supplied power to be stored and released as kinetic energy. This efficient energy transfer yields kinetic energy levels approximately 1000 times greater than those in the NC-EsMV system

(Supplementary Fig. 5). Based on the above energy landscape analysis, we further examined the equilibrium transition paths of both systems. In the NC-EsMV, the vibration follows a single monostable branch, with small positional shifts corresponding to minor equilibrium changes as the current varies between attractive and repulsive phases. In contrast, the C-EsMV exhibits bifurcation in behavior: under increasing attractive current, the magnet moves along the upper stable branch until the potential barrier diminishes and upper state loses stability, leading to an abrupt transition to the lower branch (collapse), as the current reverses, it jumps back—forming a complete hysteretic cycle (Fig. 2e). When dynamic effects are taken into account, however, the actual displacement ($z_{dynamic}$) becomes significantly larger than the static equilibrium shift, as shown in Fig. 2f. The corresponding phase portrait

confirms that C-EsMV reaches a much higher velocity and converges to a large-amplitude limit cycle compared with NC-EsMV (Fig. 2g). Finally, Fig. 2h illustrates the real-time vibration trajectory under a sinusoidal input, visualizing a single current sweep and the resulting path-dependent oscillation.

## Design principles of C-EsMV system

Our proposed mechanism offers versatility in design, relying on the balance of two dominant forces, magnetic attraction and elastic tension, emphasizing the importance of choosing suitable characteristic forces. The vibrational displacement is indeed influenced by the size of the magnets and the stiffness of the elastic membrane. However, these parameters were systematically incorporated into both our experimental design and simulations to ensure consistent scaling behavior.

To guide system design, we define characteristic scales of each force. For the characteristic elastic force ($F_{ES}^{*}$), we account for both magnet and membrane dimensions:

$$F_{ES}^{*} = \frac{\pi E}{2(1+\nu)} h_m \lambda_p^2 (R_m + R_a), \qquad (1)$$

where $E$, $\nu$, $h_m$, $R_m$, $\lambda_p$ are Young's modulus, Poisson's ratio, pre-stretched membrane thickness, membrane radius, and the prestretch ratio of the membrane, respectively. $R_a$ is the radius of the attached magnet. The characteristic magnetic force, $F_{MM}^{*}$, is defined as the maximum attractive force when the magnets are collapsed ($z = 0$):

$$F_{MM}^{*} = \frac{\mu_0}{2} M_a M_b \Omega|_{z=0}, \qquad (2)$$

where $\mu_0$ is the vacuum permeability, $M_a$ and $M_b$ are the magnetizations, and $\Omega|_{z=0}$ is a geometric factor determined by the magnet geometry. The full derivation and structural parameters are provided in Supplementary Fig. 6 and Supplementary Note 1.

By adjusting magnet thickness ($h_a$ and $h_b$) and using elastic membranes with varying moduli ($E$) while maintaining constant magnetization ($M$), we developed a design map for the maximum vibration amplitude resulting from those elastic and magnetic forces at each optimal initial position ($D_{opt}$, Fig. 3a, top). We also defined three actuation states: collapsed (magnetic force dominates, preventing the membrane from detaching the magnets touching the electromagnet), weakened (elastic force dominates, preventing magnets from retouching the electromagnet), and amplified (balanced forces allow both detachment and reattachment of magnet to electromagnet for large amplitude vibrations) (Fig. 3a, bottom). The maximum amplitude is attained along the white dashed line in Fig. 3a, where $F_{MM}^{*} \approx F_{ES}^{*}$, establishing optimal conditions for amplified vibration. Supplementary Note 4 and Supplementary Fig. 7 present additional design maps illustrating the conditions for amplification capabilities, amplifiable range ($\Delta D = D_{max} - D_{min}$), optimal initial position ($D_{opt}$), and maximum velocity ($v$), acceleration ($a$), and kinetic energy ($KE$).

To validate this design map, we experimentally examined three cases with distinct force profiles as shown in Fig. 3b–d with Supplementary Table 2: Cases I and II share the same elastic force but differ in magnetic force, while Cases II and III have the same magnetic force but different elastic forces. System states (collapsed, weakened, or amplified) depend on peak current ($I_{peak}$) and initial position ($D$) (Fig. 3b–d).

In Case I ($F_{ES}^{*} = 0.2419$ N, $F_{MM}^{*} = 0.0348$ N), weak magnetic force enables amplified vibrations (green triangles) only at short distances ($D$) or high currents. Otherwise, the system remains in the weakened state (blue circles) (Fig. 3b). In Case II ($F_{ES}^{*} = 0.2419$ N, $F_{MM}^{*} = 0.2072$ N), the magnets collapse (red dots) when $D$ is small, but achieve amplified vibrations at larger $D$ (Fig. 3c). Case III, with increased elastic force

($F_{ES}^{*} = 0.6289$ N, $F_{MM}^{*} = 0.2072$ N), exhibits amplified vibrations at small $D$ or high currents. However, at larger $D$, the system falls into a weakened state without amplification (Fig. 3d). Since Case II leverages the most significant elastic potential, it is identified as the most desirable configuration.

To quantitatively compare these three cases, we evaluated the maximum amplitude and kinetic energy at each optimal point $\alpha$, $\beta$, and $\gamma$ under the same input current ($I_{peak} = 1.5$ A). At point $\alpha$ (Case I, $D = 2.5$ mm), weak attraction keeps the magnets in a weakened state, despite being closest to the electromagnet. At points $\beta$ (Case II, $D = 4.0$ mm) and $\gamma$ (Case III, $D = 3.0$ mm), amplified vibration occurs in both cases at the boundary between amplified and weakened states, but Case II exhibits the highest performance, as experimentally measured in Fig. 3e.

To further assess the performance of Case II, we analyzed the ratio of energy conversion efficiencies ($\varepsilon_{C/NC}$, the ratio of the maximum kinetic energy of C-EsMV to that of NC-EsMV under identical electrical input) as a function of $I_{peak}$ with $D = 4.0$ mm. This ratio peaks immediately after amplification begins, reaching as high as 700, then gradually declines (Fig. 3c and f). This enables efficient, thermally stable operation of C-EsMV over time (Supplementary Fig. 8).

Conversely, by fixing $I_{peak}$ at 2.0 A and varying the distance $D$, we observed sequential transitions in system behavior (Fig. 3c and g and Supplementary Video 4). As the magnet-membrane composite approaches the electromagnet, the system moves from the weakened state to the amplified state, and finally to a collapsed state. In contrast, non-coupled systems display only monotonic changes in amplitude.

We further explored the system's behavior across varying initial position ($D$) and combinations of elastic and magnetic forces (see Supplementary Note 5 and Supplementary Figs. 9–11).

In particular, weakened states are classified into shootable and non-shootable regimes based on whether the actuator can transition into amplified state upon external triggering (Supplementary Fig. 9e, f). This triggering process, which we term "shooting," refers to a momentary stimulus that propels the magnet across the energy barrier separating the coexisting states (weakened and amplified). As shown in Supplementary Fig. 12, when the input current was increased above the onset threshold ($I_{th, on}$) and then reduced (shooting), the system exhibited a sudden transition from a weakened to an amplified vibration state, maintaining the large amplitude even after the current returned to the initial current level ($I_{peak} \approx 1.3$ A). In contrast, systems that never exceeded $I_{th, on}$ (non-shooting), remained confined to low-amplitude oscillations because the available energy was insufficient to overcome the potential barrier (i.e., the basin boundary between the two attractors). Experimentally, a brief electrical trigger induced this dynamic shift, confirming that the vibration state could be switched and retained through a transient perturbation.

This behavior was further validated through simulation (Supplementary Fig. 13). In the model, a short high-current pulse ($I_{peak} = 2.0$ A) served as the electrical trigger, driving the system across the potential barrier. Comparing the bifurcation diagrams before and after shooting at $I_{peak} = 1.4$ A revealed that the response followed a distinct path along the upper stable branch, resulting in a larger limit-cycle oscillation even in the absence of resonance. The persistence of this amplified vibration after returning to the lower input current arises from inertia, which provides residual kinetic energy to sustain oscillations around the new equilibrium branch. This inertia-driven retention is stably maintained and was further validated as an efficient energy conversion mechanism (Supplementary Fig. 14). It constitutes the vibrational hysteresis–based mechanical memory, which is further analyzed in Fig. 5.

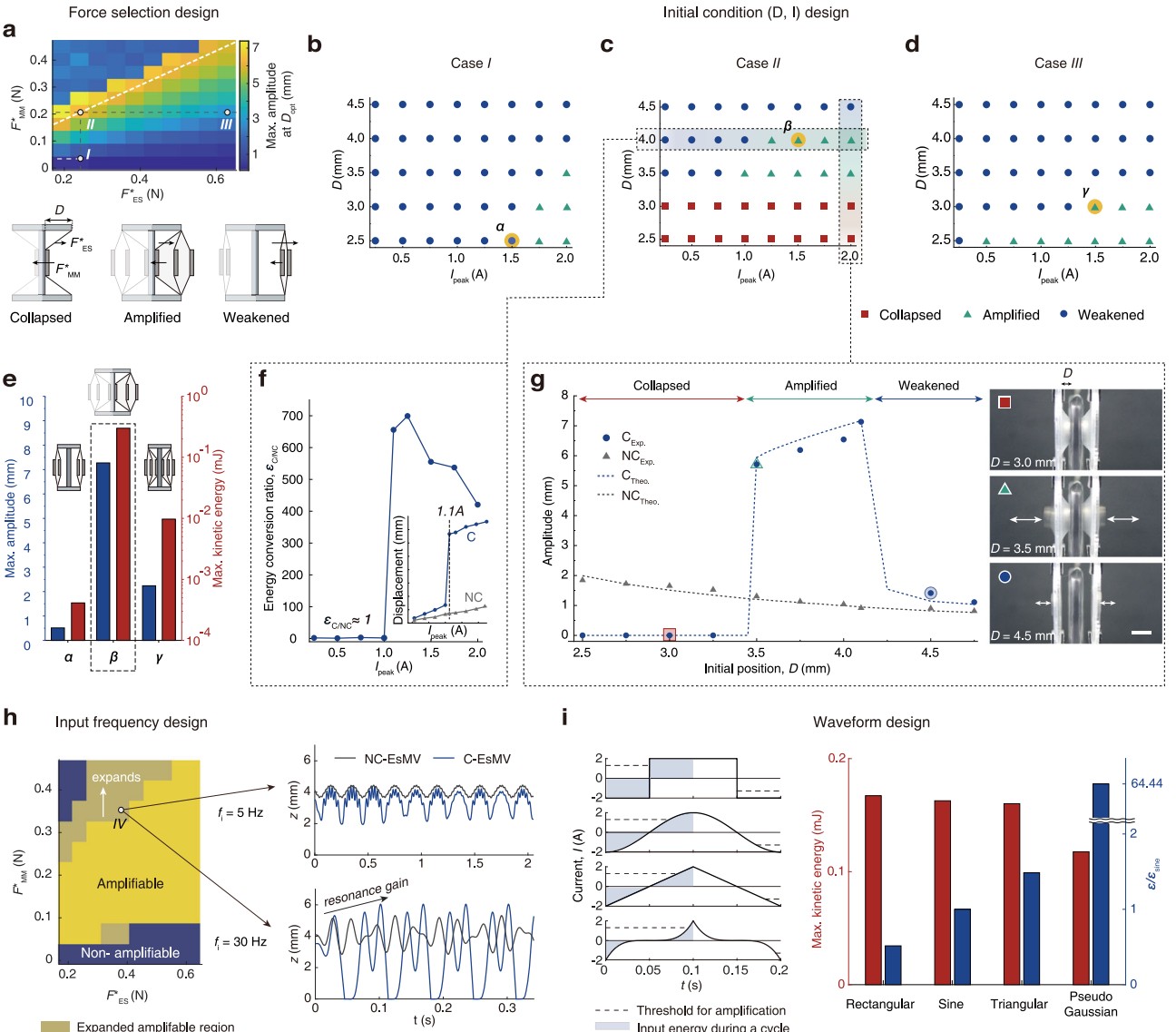

**Fig. 3 | Design principles of EsMV system. a** Schematic illustration of three states of C-EsMV system (*Collapsed, Weakened, and Amplified*) and a design map based on the selection of materials with characteristic forces: magnetic force, $F^*_{MM}$, and elastic force, $F^*_{ES}$. **b–d** State maps as a function of peak input current ($I_{peak}$) and initial position ($D$) in three cases selected from the design map: **b** *Case I*, **c** *Case II*, and **d** *Case III*. Magnetization ($M$) is identical across all systems, with forces controlled by the elastic modulus of the membrane ($E$), and the magnet thickness ($h$). **e** Maximum displacement and kinetic energy under optimized conditions for each case ($\alpha$, $\beta$, $\gamma$) at the same peak current ($I_{peak} = 1.5$ A). **f** Energy ratio ($\varepsilon_{C/NC}$) between NC-EsMV and C-EsMV systems as a function of $I_{peak}$ with a fixed $D$ (4.0 mm).

**g** Amplitude as a function of $D$ at constant $I_{peak}$ (2.0 A) in each system. In C-EsMV, amplitude is divided into three distinct states unlike NC-EsMV, as shown in the snapshots. Dots represent experimental data, and dashed lines indicate simulations. Scale bar, 5 mm. **h** Effect of input frequencies. At $f_i = 30$ Hz, systems near resonance transition from the weakened state to the amplified state, expanding the amplifiable region. **i** Effect of waveform types on actuation efficiency. As the input signal becomes more concave, the required input energy for amplification is significantly reduced, greatly improving efficiency compared to sine waves (~64 times).

## Frequency effect on system design

While a higher input current magnitude can extend the range over which amplification occurs (Supplementary Fig. 15), the associated increase in energy consumption constrains its practical applicability. An alternative strategy is to tune the input frequency to the membrane's natural frequency, taking advantage of near-resonance behavior (Fig. 3h). Under near-resonant conditions, some systems that previously exhibited weakened vibrations—which could not be amplified, even at optimal initial position—transition to amplified vibrations, expanding the amplifiable range ($\Delta D$). For example, *Case IV* ($F^*_{ES} = 0.3870$ N, $F^*_{MM} = 0.3463$ N), indicated in Fig. 3h, behaves similarly to NC-EsMV at 5 Hz but switches to an amplified state when the input frequency approaches the membrane's natural frequency, 30 Hz,

due to resonance gain (Supplementary Fig. 16). It is worth noting that this resonance-induced amplification strongly depends on the viscoelasticity of the membrane. As shown in Supplementary Fig. 17, at the dynamic viscosity used in our experiments ($\eta = 50$ Pa·s, $\eta/E = 0.001$ s), a system that exhibited weakened vibration at 5 Hz transitioned to a stable, resonance-driven amplified vibration at 30 Hz. When the dynamic viscosity was increased by one or more orders of magnitude, the amplification effect progressively diminished, and at high damping ($\eta = 50,000$ Pa·s, $\eta/E = 1$ s) the resonance response nearly disappeared.

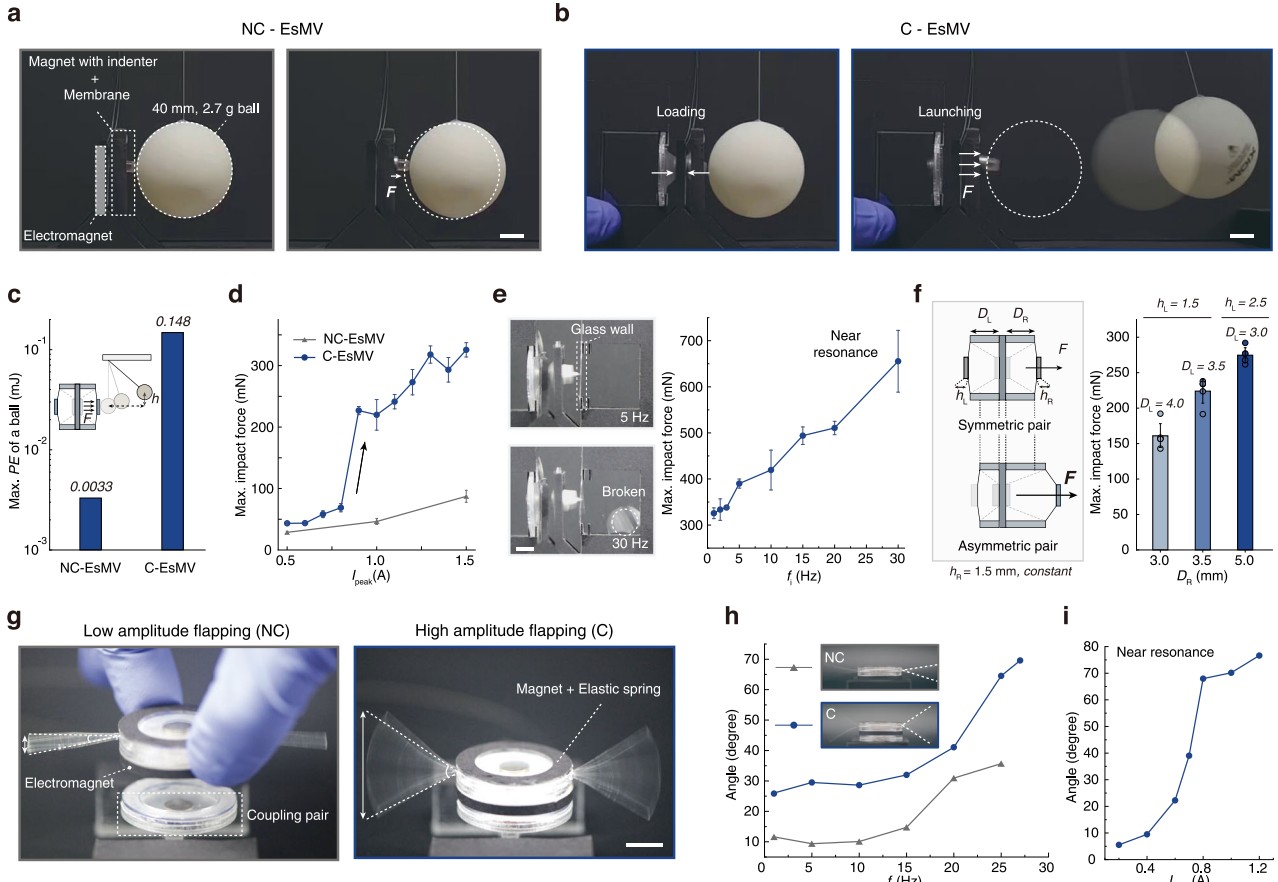

**Fig. 4 | Amplification of mechanical forces and displacements in EsMV actuators. a**, **b** Launching a ball in each system, **a** NC-EsMV and **b** C-EsMV. Despite identical energy input ($I_{peak}$ = 1.0 A), greater elastic energy stored in the membrane results in a more forceful launch in the coupled system (Scale bar = 1 cm). **c** Maximum potential energy of the ball in two systems. **d** Maximum impact force as a function of input current in each system ($f_i$ = 1 Hz). **e** Input frequencies effect on impact forces. As the frequency approaches the resonance, maximum impact force is increased, leading to a breaking of glass (0.1 mm) at higher frequencies ($f_i$ = 30 Hz, $I_{peak}$ = 1.5 A). Scale bar, 5 mm. **f** Geometric asymmetry effect on impact forces. As the initial position on the right side ($D_R$) between the magnet and electromagnet increases, controlling the left position ($D_L$) and the thickness of magnets ($h_L$, $h_R$) modulates vibration amplifications. **g** The flutter of wings with a NC-EsMV actuator and a C-EsMV actuator. Scale bar, 1 cm. **h** The angle of the flapping as a function of frequency in each system. Inset snapshots show the motion of flapping at each system near its resonance frequency ($f_{i, NC}$ = 25 Hz, $f_{i, C}$ = 27 Hz). **i** The angle of the flapping as a function of input current in C-EsMV actuator operating near resonance ($f_{i, C}$ = 27 Hz). In all experiments, error bars denote SDs; $n$ = 5.

## Designing waveforms for system efficiency

The energy conversion efficiency of the system—defined as the ratio of the maximum kinetic energy of the vibrating magnet to the supplied electrical energy—is relatively low (<1%) because of the limited coil turns and associated geometric factors of the electromagnet. However, this efficiency can be substantially enhanced by waveform design (Supplementary Note 6 and Fig. 3i), as the waveform shape directly affects the input energy required for amplified vibrations. We numerically simulated a wide range of waveforms, spanning from convex (rectangular) to concave (pseudo-gaussian) upward. Convex-shaped waveforms facilitate rapid electromagnetic repulsion, with rectangular waveforms achieving the earliest take-off and longest duration before landing (see Supplementary Fig. 18). In contrast, concave-shaped waveforms delay the current rise to the threshold, postponing the onset of take-off. However, as long as the peak current exceeds the threshold, amplification occurs regardless of waveform shape. Notably, concave waveforms, particularly pseudo-gaussian type, offer significantly higher energy conversion efficiency because of reduced energy consumption. Simulations show that a pseudo-gaussian waveform can improve efficiency by roughly 64.4 times compared to a sine wave ($\varepsilon/\varepsilon_{sine}$), confirming the potential for more efficient operation through deliberate input waveform design.

## Amplification in force and displacement

We demonstrate the utility of this mechanism in amplifying both force and displacement. We created a slingshot-like system by attaching an indenter, serving as a hammerhead, to the vibrating magnet to strike a target object. A ball (40 mm in diameter, 2.7 g in mass) was placed in contact with the indenter (Fig. 4a, b). Under NC-EsMV, limited energy transfer resulted in minimal displacement. In contrast, C-EsMV rapidly transferred energy to the ball, utilizing the nonlinear, stepwise release mechanism, propelling it substantially farther. Consequently, the ball's maximum potential energy ($P.E. = mgh$) jumped from 0.0033 mJ (NC-EsMV) to 0.148 mJ (C-EsMV)—a 50-fold improvement (Fig. 4c).

To quantify impact force, we measured the force exerted by the vibrating magnet (Supplementary Fig. 19). NC-EsMV produced a relatively small force ($F_{peak}$ ~ 87 mN at $I_{peak}$ = 1.5 A) that scaled linearly with the input current. C-EsMV showed a nonlinear response, with the force rising from 68 mN at $I_{peak}$ = 0.8 A to over 225 mN at $I_{peak}$ = 0.9 A (1 Hz, Fig. 4d), releasing stored elastic energy at a high rate over a short period, resulting in high power output. Operating near resonance further magnified the impact force as the frequency varies from 1 to 30 Hz at a fixed current ($I_{peak}$ = 1.5 A), the maximum impact force increased from approximately 300 mN to 650 mN (Fig. 4e). At lower frequencies, the force was insufficient to break a thin glass wall

(0.1 mm thickness), but at 30 Hz, the vibrating magnet fitted with a beak-shaped indenter successfully shattered the glass wall. Beyond resonance, vibration amplitude and impact force diminished markedly.

So far, we have presented C-EsMV systems with symmetric magnet-membrane coupling. Asymmetric vibrations can also be realized by altering system geometry, material properties ($M$, $E$), or magnet's initial position ($D$) (Supplementary Fig. 20). Such asymmetry leads to an unequal distribution of elastic potential energy, influencing both the amplifiable range and the amplitude. Increasing distance $D_R$ on one side under the same energy input ($I_{peak} = 1.0$ A) asymmetrically raises stored elastic energy, thereby boosting the maximum impact force and demonstrating the system's adaptability in tuning impact dynamics (Fig. 4f). To maintain overall balance, the opposite side was adjusted by increasing its magnet thickness and reducing its gap, compensating for the weakened magnetic attraction. Each configuration was tested as an independent system, confirming that asymmetric coupling can effectively localize and enhance actuation on a chosen side.

Beyond force amplification, the amplified vibration can be converted to large fluttering motions (Fig. 4g). By attaching wings (PET, 0.1 mm thickness) to the magnet in a lever-like configuration (Supplementary Fig. 21), the C-EsMV system achieved significantly larger wing flaps than NC-EsMV system, quantified by angular displacement ($\theta$) from 1 Hz to near resonant frequency, 27 Hz (Fig. 4h and Supplementary Video 6). Even under near resonant conditions, the flapping amplitude of C-EsMV exhibited a nonlinear response to the applied current (Fig. 4i).

## Mechanical memory

The ability of mechanical systems to autonomously respond to external stimuli, exhibiting embodied physical intelligence without the need for programmed controls, is of great interest[42,43]. Building on the nonlinear hysteretic behavior and the "shooting" phenomenon of C-EsMV, we introduce a concept of mechanical memory with exteroceptive characteristics (Fig. 5a) requiring no complex manipulation. By replacing the rigid spacer with an elastic spacer, which can deform and recover when subjected to external triggers, the system transitions from a weakened vibration state (*standby*) to an amplified vibration state (*memorized*) whenever the spacer is compressed, drawing the magnets closer and producing a "mechanical shooting" effect (Supplementary Fig. 22). This process is analogous to the electrical shooting described earlier, relying on inertia–elastic energy exchange that drives the transition from the weakened to the amplified vibration state (Supplementary Fig. 23). Once triggered, the system sustains oscillation within the amplified branch even after the external input is removed. As shown in Supplementary Fig. 24, this design effectively stores external mechanical stimuli as persistent amplified vibrational states—a phenomenon we term mechanical memory.

We characterize this mechanical memory using four regimes (*R1, R2, R3,* and *R4*) defined by characteristic currents on a current-amplitude hysteresis graph (Fig. 5b): $I_{th, off}$, $I_{ext.cept}$, and $I_{th, on}$. As shown in Fig. 5c, d, Regime 1 (*R1*, $I < I_{th, off}$) requires significant trigger force and displacement for amplified vibration, but immediately reverts to *standby* when the trigger is removed, preventing memorization. Regimes 2 (*R2*) and 3 (*R3*) lie within the exteroceptive range, each exhibiting distinct memory behaviors. In *R2* ($I_{th, off} < I < I_{ext.cept}$), memory retention is positively correlated with the trigger duration, creating a "volatile memory mode" where *memorized* states eventually reset. In *R3* ($I_{ext.cept} < I < I_{th, on}$), brief mechanical triggers induce permanent amplified vibration, defining a "non-volatile memory mode". Regime 4 (*R4*, $I > I_{th, on}$) allows amplified vibration without any trigger (Fig. 5c, Supplementary Fig. 25, and Supplementary Video 6).

We evaluated the trigger force ($F_T$) and trigger displacement ($D_T$) required for activation (Fig. 5e, corresponding to Supplementary

Fig. 9e). At currents above $I_{th, on}$ (*R4*), $F_T$ and $D_T$ are zero because amplification occurs spontaneously. As the current decreases further below $I_{th, on}$ (*R3 ~ R1*), greater trigger force and displacement are needed. For example, at $I_{peak} = 1.5$ A, the system can be activated with a minimal force ($F_T \approx 0.17$ N) and displacement ($D_T \approx 0.5$ mm). Consequently, modulating $I_{peak}$ allows tuning of trigger sensitivity, while varying the modulus of the elastic spacer provides additional control over the activation requirements (Supplementary Fig. 26).

To assess the longevity of the triggered amplification after the trigger is removed, we measured retention time ($t_R$) (Fig. 5f). In *R4*, amplification is perpetual and does not depend on a trigger. In *R3*, even a brief trigger (trigger time, $t_T < 0.1$ s) induces indefinite amplified vibration, demonstrating non-volatile memory. However, the duration of this non-volatile retention is governed by the viscoelastic property of the membrane (Supplementary Fig. 27): higher viscosity increases damping and gradually suppresses sustained oscillations, whereas a more elastic membrane maintains stable amplified vibration. This confirms that membrane elasticity is essential for robust memory retention. In *R2*, retention depends on how long the trigger is applied—short triggers ($t_T = 0.1$ s) cause rapid decay, while relatively longer triggers ($t_T = 10$ s) extend retention to nearly 6 s. This volatile nature arises from membrane resonance gains and the elastic spacer's relaxation time, enabling tunable memory retention. In *R1*, the system immediately returns to *standby* once the trigger is removed.

To visualize mechanical memory as a trigger tracer, we built a $3 \times 3$ array of independent C-EsMV units (Fig. 5g). In the volatile mode, applying a 15 s trigger to one cell (C1) and a 7 s trigger to another (C9) yields retention times of 7 s and 1.5 s, respectively. (Fig. 5h, top). In the non-volatile mode, cells were sequentially activated in a programmed pattern, maintaining amplified vibration indefinitely (Fig. 5h, bottom and Supplementary Video 7). The system can be reset through external interventions, such as touching the vibrating magnet to dissipate its kinetic energy (Supplementary Fig. 28).

Since the memory effect relies on bistable, hysteretic response of the membrane-magnet composites, applying force directly to the magnet rather than the spacer produces the same outcome (Supplementary Fig. 29). In addition to the non-contact trigger-based memory described earlier (Fig. 1h), any mechanical trigger that gives additional elastic energy can utilize this mechanism, broadening the potential applications of this mechanical memory across diverse environments.

## Discussion

The proposed C-EsMV system leverages magnetic attraction and elastic regulation to induce Elasto-Magnetic Instability (EsMI), enabling amplified inertial motion through a simple and scalable mechanism. This dynamic collapse, governed by the interplay between elastic and magnetic forces, leads to vigorous vibrations and efficient energy release, achieving up to a 700-fold increase in energy efficiency relative to the NC-EsMV system.

By systematically tuning magnetic and elastic parameters, we establish design principles—summarized in the design map—that generalize across different force scales. Inertia-driven dynamics enhance two key features: nonlinear response and hysteretic behavior, enabling stepwise force and motion output. These behaviors are demonstrated in applications such as slingshot-like actuation, wing flutter, and mechanical memory with either volatile or non-volatile characteristics.

While conventional electromagnetic actuators are designed to maximize force and speed, their efficiency typically declines at high currents, limiting their integration into compact systems. In contrast, the C-EsMV system offers an alternative approach that enables energy-efficient, stepwise responses suitable for space- and power-constrained environments. Moreover, because the underlying instability mechanism is not limited to electromagnetic control, it may be extended to other forms of actuation, including pneumatic and

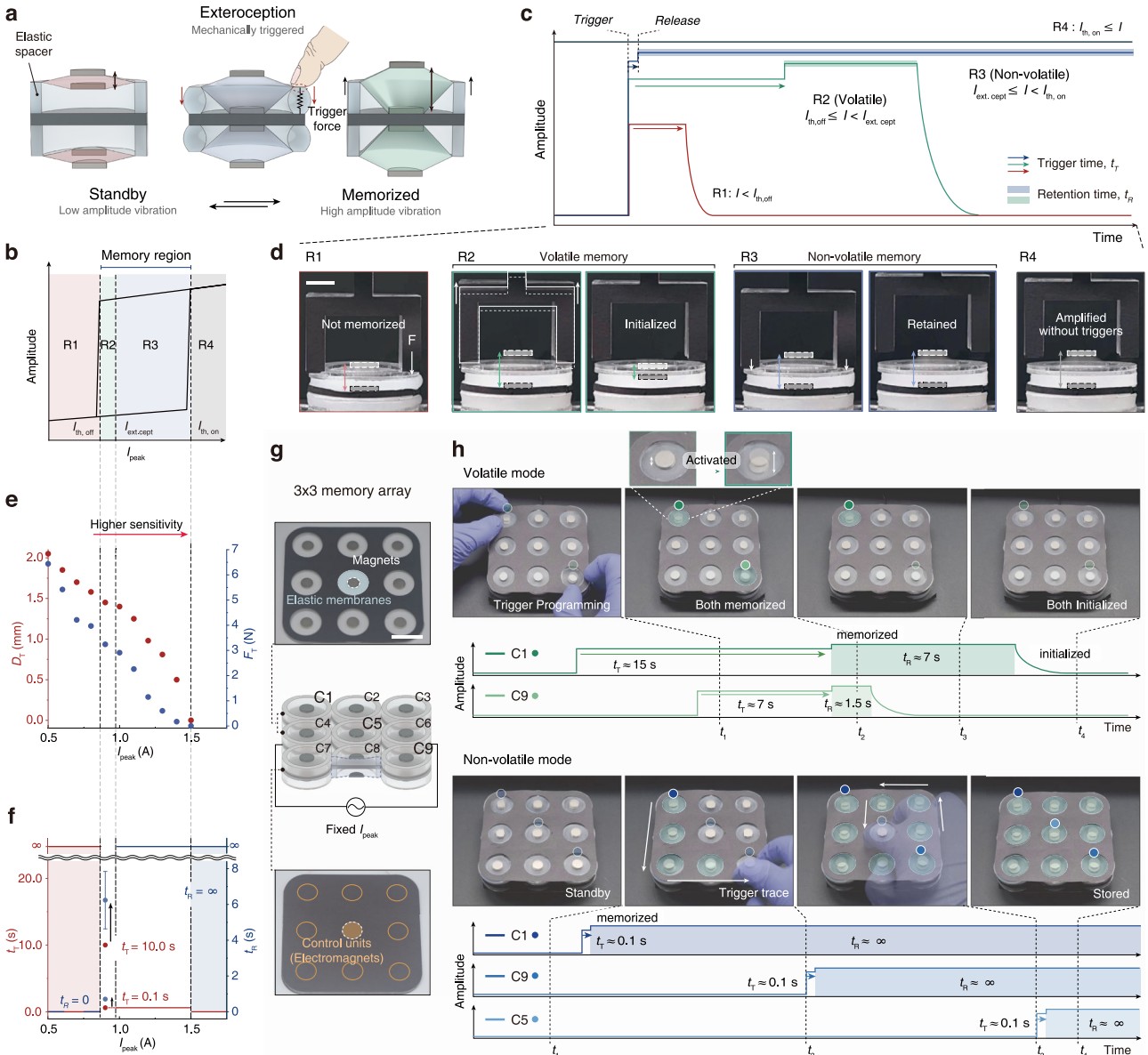

**Fig. 5 | Volatile and non-volatile memories operated by a mechanical triggering. a** Schematic illustration of mechanical memory using C-EsMV. **b–d** Behavior of mechanical memory: **b** Schematic graph of hysteretic motion as a function of current, with the system divided into four regimes based on three characteristic currents ($I_{th, off}$, $I_{ext.cept}$, $I_{th, on}$). **c** Vibration amplitude over time in each regime. In *R1* ($I < I_{th, off}$) and *R2* ($I_{th, off} \le I < I_{ext.cept}$), the system exhibits volatility, with retention possible only in *R2*. In *R3* ($I_{ext.cept} \le I < I_{th, on}$), non-volatility is observed, maintaining amplified vibration indefinitely after triggering. In *R4* ($I_{th, on} \le I$), amplification occurs without a trigger. **d** Representative vibration behaviors in each region. Scale bar, 8 mm. **e** Trigger force ($F_T$) and displacement ($D_T$) required for memorization in each regime. **f** Retention time ($t_R$), indicating how long the system retains the amplified vibration after the trigger is removed. It is shown as a function of trigger duration ($t_T$) in each regime. Error bars denote SDs; $n = 5$. **g** Structure of $3 \times 3$ array units for a mechanical trigger "tracer" composed of elastomer membranes, coupled magnets, and electromagnets (Scale bar = 2 cm). **h** Function of volatile and non-volatile memory modes over time based on the amplitude. Volatile mode operates at $I_{peak} = 0.9$ A in *R2*, while non-volatile mode is activated at $I_{peak} = 1.1$ A in *R3*. Both functions are demonstrated at $f_i = 30$ Hz.

chemical triggers. The demonstrated ability of the C-EsMV system to achieve nonlinear amplification and motion hysteresis highlights a broader design framework for adaptive mechanical systems. This approach may be particularly suited for scenarios that require discrete, energy-efficient responses to small stimuli with programmable thresholds—such as mechanical transistors, spike-signal processors, or memory-integrated devices. More broadly, these finding suggest that embedding inertia as an active design element in instability-driven architectures provides a distinct and versatile route toward programmable, efficient, and compact actuation systems.

## Methods

### Fabrication of EsMV system

Both NC-EsMV and C-EsMV systems utilized identical NdFeB permanent magnets (Jungsin Corp.) and electromagnets (TDK Corp., WR151580-48F2-G). The permanent magnets (radius = 4 mm, thickness = 0.5 mm, three-layer stack) exhibit a typical magnetization of $M_a = M_b = 4 \times 10^5$ A/m, which was experimentally verified by measuring the magnetic attraction force as a function of distance. This magnetization value was consistently used in both experiments and simulations. Elastic membranes were custom-fabricated using spin coating (Dong Ah Trade Corp, ACE-200) on a Teflon substrate to ensure uniform thickness. Coated elastomers were cured at 60 °C for 1 h.

Membranes were pre-stretched using an acrylic frame at specified stretch ratios ($\lambda = 1.5$) to adjust elastic forces. Donut-shaped acrylic spacers, cut with a laser cutter (Universal Laser System, VLS3.50), were attached to the pre-stretched membranes to control the distance between the electromagnet and the magnet, i.e., initial position $D$. In NC-EsMV, the assembly (magnet + membrane + spacer) was mounted on one side of the electromagnet. For C-EsMV system, an identical assembly was symmetrically added to the opposite side of the electromagnet.

### Measurement of the real-time position of the vibrating magnet
To analyze magnet motion during vibration, an acrylic-based indenter (0.1 g) with a red marker was attached to the vibrating magnet. Motion was recorded using a high-speed camera (Phantom, V611-32G-MAG) at various driving frequencies. Magnet position was tracked using custom MATLAB code, and the amplitude, velocity was obtained by differentiating the position data with respect to time.

### Characterization of elastic membranes and magnets, and basic force measurement
Three elastomer types with varying moduli (Smooth-On Inc., EcoFlex 0020, EcoFlex 0050, Dragon Skin NV10) were used. Elastomers were prepared by mixing parts A and B in a 1:1 ratio per supplier instructions. Membranes were pre-stretched and tested for elastic restoration forces by pressing the center of the membrane using a universal testing machine (Instron 3343). Magnetic forces were controlled by stacking magnets (0.5 mm thick, 8 mm diameter). Magnet-magnet forces were measured by fixing one magnet to the base of the testing machine and recording the force required to detach another magnet. Similarly, electromagnet-magnet forces were measured by fixing the electromagnet to the base and recording forces at various currents. Key parameters, including $E$, $c$, $M_a$, and $M_b$, were extracted through the experiments and applied to the analytical model. The predicted force profiles closely matched the measured results (Supplementary Fig. 2b–d).

### Construction of state regime map for various magnetic and elastic conditions
The electromagnet was fixed in the center, with a linear stage (Pinetek, PX30-CL15) on each side. Each stage held a pre-stretched membrane and an attached magnet, enabling precise control of initial position ($D$) between the electromagnet and the magnet-membrane pairs. This setup allowed identification of system's three states—*collapsed*, *amplified*, and *weakened*—represented in state regime map (Fig. 3b–d).

### Electrical input source characterization
To drive the EsMV systems, a function generator (Agilent, 33612A) was used to produce various waveforms (sine/rectangular/triangular). A current amplifier (ACCEL Instruments, TS250-0) or DC power supply (Keysight, E36233A), combined with custom circuit amplifier (OPA541), delivered the required current for experiments.

### Measurement of trigger force and displacement in mechanical memory
Elastic spacers were fabricated using molding processes with elastomers of different moduli (EcoFlex 0020 and Elite Double, Zhermack) (Supplementary Fig. 26). Trigger force, trigger displacement (the pressing distance needed for amplification), and retention time (the duration of amplified vibration after trigger) were quantified using a universal testing machine. This setup controlled pushing speed, hold time, and retraction speed, enabling measurements across different mechanical memory regimes (Supplementary Fig. 25).

## Data availability
All data supporting the findings of this study are provided with the paper, including in the Source Data and Supplementary Information. Source data are provided with this paper.

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

## Acknowledgements

This work was supported by a National Research Foundation of Korea (NRF) grant funded by the Korean Government (No. RS-2024-00459269 and 2018-052541).

## Author contributions

S.-Y.C., J.-S.P., and J.-Y.S. conceived the idea and wrote the manuscript. S.-Y.C. and J.-S.P. designed, conducted, analyzed the experiments. J.-S.P. developed the theoretical modeling and performed simulations. W.J.S., M.K., Y.H.L., Y.E.C., and H.L. supported demonstrations and video recordings. All authors discussed the results and commented on the manuscript. J.-Y.S. and H.-Y.K. supervised the study.

## Competing interests

The authors declare no competing interests.
