## [Transparent Peer Review file · Nature Communications]

Elasto-magnetic instabilities for amplified actuation and mechanical memory

Corresponding Author: Professor Jeong-Yun Sun

Version 0:

Reviewer comments:

Reviewer #1

(Remarks to the Author)

This paper presents some interesting experimental demonstrations of some intriguing systems, that exploit the interplay between repelling and attracting forces to create a complex energy landscape with some highly complex responses. There is good and interesting work here, however I do not believe the paper is of sufficient quality for publication in nature communications. My main reason for this conclusion, is that the authors have described a nonlinear and dynamic system, but seem to have completely ignored the science of nonlinear dynamics in describing it. Many of the interesting phenomena described here have are widely understood in other systems; for example what the authors describe as a “memory” effect is really the ability of nonlinear oscillators to show multiple steady state responses to the same excitation depending on their initial conditions, and therefore disturbances can cause the active response to change and remain dynamically stable. The effect described as ‘analogous to a Schmitt trigger’ is what is known as a bifurcation, whereby systems exhibit substantial changes to relatively minor changes to parameters, and has an extensive and well documented theory. I would urge the authors to reexamine their work in the light of nonlinear dynamics theory, as it will help them to explain it more effectively and for them to understand where their true potential for novelty lies.

To add a little further to the above point, the authors do make general references to the role of inertia in some of the behaviours shown and state that ‘Despite the inherently dynamic nature of instabilities, most existing systems assume quasi-static behaviors, overlooking the fundamental role of inertia during fast state transitions.’ However, structural dynamics explicitly includes inertia in equations of motion as a matter of course, and has been applied to substantially nonlinear systems including those with instability and multistability.

I also think some of the claims for energy efficiency and ‘amplifying kinetic energy’ need to be stated with a great degree of care - physics tells us that energy is never created, however an oscillating system can contain a large amount of conserved energy that it exchanges between kinetic and potential energy during oscillation, and these oscillations can therefore be sustained with very little ongoing energy input. So to imply that the system is giving us any energy for free is simply wrong, however I think examples such as striking the suspended ball do reveal that exploiting instabilities in these ways can allow energy to be released at a very high rate, ie. it is accurate to describe these systems as potentially showing very high power over short periods.

Despite this negative review I do encourage the authors to continue the work and am sure that many publications will ensue. The devices certainly capture the imagination, in particular for the suggested applications of a flutter actuator and a ‘mechanical transistor’.

Reviewer #2

(Remarks to the Author)

This manuscript reports an elasto-magnetic instabilities that can enable gigantic actuations and mechanical memory. The authors have designed a coupled elasto-magnetic vibration system with a pair of magnets and elastic membrane. Both the actuated force and displacement are much higher than a non-coupled control system. The work is original in working principle and the results of mechanical memory are impressive. The data shown in main text and extended data can support the conclusions. Thus, I recommend the publication of this manuscript after minor revision. The following issues need to be addressed by the authors:

(1) In the manuscript, the distance between the magnet and coil D is carefully investigated as a parameter. In the design of the coupling system, will the size of the magnet and the elastic membrane affect the vibrational displacement? What is the

intensity of magnetization for the magnet?

(2) Why do both systems exhibit a rapid decay in vibration amplitude beyond resonance (Line 104)? Is there any high-order resonance frequency for the system?

(3) The demo of mechanical memory in Fig. 5 is interesting. More description for the working mechanism is needed. The guess is that the hysteric behavior in the plot of Amplitude-I_{peak} is key to the memory behavior. Why is there a hysteresis in this curve?

(4) In the R3 regime of mechanical memory, the system is non-volatile. The elastic membrane is vibrating continuously. Will the viscoelasticity of the polymeric membrane affect the vibration and mechanical memory?

Reviewer #3

(Remarks to the Author)

This manuscript presents a very interesting and useful electromagnetically driven actuator system. The authors introduce an Elasto-Magnetic Vibration (C-EsMV) mechanism that leverages inertia-driven dynamics and modulated bistability arising from the interplay between magnetic attraction and elastic tension, thereby amplifying both force output and displacement. Furthermore, the concept of inertial vibration hysteresis is used to demonstrate different forms of mechanical memory with tunable performance. Overall, the system is well designed, the concept is convincing, and the data and analysis are carried out in a systematic manner. This work deserves publication in Nature Communications.

However, the manuscript seems to have been structured with Nature in mind, which results in the writing and discussion in the main text being rather condensed. As a consequence, much of the essential data has been placed in the Extended Data and Supplementary Figures, making the overall discussion less easy to follow. In addition, some figures contain a large amount of information but only limited explanation, which makes interpretation less straightforward. I would suggest the authors revise the manuscript by expanding the discussion and improving clarity.

Specific comments:

1. Understanding of the instability. On line 70, the authors describe: "The resulting combination of magnetic and elastic force generates an elasto-magnetic instability (EsMI) with two stable states (Figure 1d)." Since the electromagnet under AC current can show "off, attract, repulse" states, how should one understand the concept of two stable states in this multi-state context? It seems that the authors use the static equilibrium (off state) as the reference, but the relation to instability needs clearer explanation.
2. The stepwise response and large-amplitude, high-efficiency actuation of C-EsMV compared to NC-EsMV are clearly shown in Figure 1e,f. However, if the input current is further increased, will the maximum amplitude of NC-EsMV eventually approach that of C-EsMV, since the contribution of the coupled magnets is no longer the dominant factor in the overall magnetic force? This is consistent with Figure 1f, in which the amplitude difference between C-EsMV and NC-EsMV becomes smaller near resonance.
3. The manuscript repeatedly discusses energy efficiency, but there is no clear definition or description of how it is calculated. This should be explicitly provided in the main text.
4. Regarding Figure 4f (right side, asymmetric vibration), were DR, DL, and HL all changed simultaneously? Were only three separate groups tested? Please clarify.
5. What exactly is meant by the membrane's natural frequency? Will this change when the modulus or other parameters of the membrane are varied?
6. Figure 5, which presents four memory regions, is highly interesting. However, the figure contains too much information while the mechanistic discussion is too limited, making it difficult to clearly understand the results.

Minor comments:

7. Some figures and captions are unclear, and the authors are encouraged to review and improve them. for example:
 - a) in Figure 1c and d, what do the inserted physical photos represent?
 - b) In Figure 2, the font size is too small and the grey color makes it hard to read. The many overlapping lines obscure the key point (e.g., inertial under/overshoot).
 - c) In Figure 3i, the vertical axis " η/η_{sine} " needs definition.
8. Numerical formatting should be consistent: e.g., line 306 states "70,000% increase in energy efficiency," whereas line 195 states "700"; please unify style.
9. It is suggested that recent related work be cited, for example:
Y. Tian, Adv. Funct. Mater. (2025), "A Dynamically Programmable Hydrogel Surface with Rapid Magnetically Actuated Snapping of Bistable Dome Configurations" (doi:10.1002/adfm.202508885).

Version 1:

Reviewer comments:

Reviewer #1

(Remarks to the Author)

The authors have responded well to my comments. This was already an interesting and novel paper, and now it has much stronger links to relevant fields of academic literature. I recommend acceptance.

Reviewer #2

(Remarks to the Author)

The authors have done very careful revision. I recommend the publication of this manuscript.

Reviewer #3

(Remarks to the Author)

My concerns have been well addressed by the authors. Now the manuscript is comprehensive. It is a carefully done work. An acceptance for publication is recommended.

Response to reviewers

We sincerely appreciate the reviewer's valuable comments on our manuscript. Below, we provide point-by-point responses to the reviewer's comments and the corresponding changes in the revised manuscript. Sentences revised are marked in red.

Reviewer #1

Remarks to the Author: This paper presents some interesting experimental demonstrations of some intriguing systems, that exploit the interplay between repelling and attracting forces to create a complex energy landscape with some highly complex responses.

Response: We thank the reviewer for the time and effort devoted to evaluating our manuscript. We have carefully considered all comments. Below, we address each point and explain the corresponding revisions made to the manuscript.

Comment: 1. There is good and interesting work here, however I do not believe the paper is of sufficient quality for publication in nature communications. My main reason for this conclusion, is that the authors have described a nonlinear and dynamic system, but seem to have completely ignored the science of nonlinear dynamics in describing it.

Response: We sincerely thank the reviewer for this thoughtful and constructive assessment. We appreciate that the system and experimental demonstrations were found to be interesting, and we fully acknowledge that our original framing did not sufficiently connect the observed behaviors to the established nonlinear dynamics framework. As correctly noted, features such as multistability, bifurcation-like thresholds, and hysteresis are classical aspects of nonlinear structural dynamics.

In the revised manuscript, we have extensively reinterpreted our system within this framework. Specifically, we now:

- Analyze the behaviors of the elasto-magnetic vibration (EsMV) and mechanical memory systems using standard concepts of **bifurcation**, **multistability**, and **path-dependent switching**;
- Include **phase-portrait** analyses that reveal coexisting stable vibration states and trigger-induced transitions between them; and
- Add citations and discussions linking our results to prior theoretical and experimental studies that incorporate inertia in nonlinear dynamics.

At the same time, we clarify that the novelty of our work does not lie in redefining these classical concepts, but in harnessing them within a tunable elasto-magnetic architecture. By treating inertia as a controllable design parameter, we demonstrate how instability-driven dynamics can be programmed and utilized for actuation and memory in soft materials. These revisions strengthen the theoretical foundation of our study and align our discussion with the reviewer's valuable recommendation.

Comment: 2. Many of the interesting phenomena described here have are widely understood in other systems; for example what the authors describe as a “memory” effect is really the ability of nonlinear oscillators to show multiple steady state responses to the same excitation depending on their initial conditions, and therefore disturbances can cause the active response to change and remain dynamically stable.

The effect described as ‘analogous to a Schmitt trigger’ is what is known as a bifurcation, whereby systems exhibit substantial changes to relatively minor changes to parameters, and has an extensive and well documented theory.

I would urge the authors to reexamine their work in the light of nonlinear dynamics theory, as it will help them to explain it more effectively and for them to understand where their true potential for novelty lies.

Response: We thank the reviewer for this insightful and important comment. We fully agree that our system is naturally interpreted within the framework of nonlinear dynamics, which is essential for both clarity and novelty. Our aim is to bridge the theoretical framework with practical implementations in soft actuation. Building on the reviewer's valuable comment, we reinterpret our system using classical nonlinear structural dynamics and address the points in three parts.

1) Reexamination of Elasto-Magnetic Vibration (EsMV) system via nonlinear structural dynamics.

[1.1. Bifurcation diagram and vibration behaviors with phase-portraits]

The elasto-magnetic instability (EsMI) in the coupled EsMV (C-EsMV) system arises from the balance between magnetic attraction and elastic restoring force and manifests as bifurcation—small parameter changes cause large transitions between stable responses. To provide a closer analysis, we examined both the NC-EsMV and C-EsMV in terms of equilibrium positions and energy landscapes as functions of current (Fig. R1).

For the NC-EsMV, starting from an initial position (D), the system remains monostable under both attractive (-2.0 A) and repulsive ($+2.0$ A) electromagnetic forces. In contrast, the C-EsMV reveals a bistable region as the current is gradually varied from attractive to repulsive, with pronounced bifurcations identified at Point A (saddle-node fold) and Point B (subcritical Hopf).

Fig. R1. (a) Schematic of the NC-EsMV with variable current (I). (b) State diagram showing equilibrium positions of the NC-EsMV system as a function of input current. (c) Energy landscape of the NC-EsMV system plotted as a function of position for different input currents (-2 , 0 , and 2 A). (d) Schematic of the C-EsMV. (e) State diagram showing equilibrium positions of the C-EsMV system as a function of input current. (f) Energy landscape of the C-EsMV system plotted as a function of position for different input currents (-2 , 0 , and 2 A) (Point A: Saddle-node fold, Point B: Subcritical Hopf).

To decouple resonance effects, we fixed the input frequency at $f_i = 5$ Hz and compared the bifurcation paths, vibration amplitudes, and phase portraits (Fig. R2). Below the threshold current ($I_{\text{peak}} = 1.4$ A), the C-EsMV cannot

cross the unstable energy barrier; thus both systems exhibit low-amplitude/low-velocity oscillations (Fig. R2a, b). Above the threshold ($I_{\text{peak}} = 2.0$ A), the C-EsMV crosses the barrier and shows amplified high-amplitude/ high-velocity oscillation (Fig. R2c, d).

These behaviors share strong analogies with Duffing-type oscillators, where amplitude-dependent resonance, hysteresis, and jump phenomena arise from nonlinear stiffness. The resulting bifurcation structure, together with inertia, accounts for the mechanical memory effects discussed later. We have incorporated this interpretation into the revised manuscript and Supplementary materials.

Fig. R2. Comparison of current-dependent vibration behaviors in NC-EsMV and C-EsMV systems based on bifurcation analysis. (a, c) Static analysis based on equilibrium positions showing motion trajectories at (a) $I = \pm 1.4$ A and (c) $I = \pm 2.0$ A for NC-EsMV (gray dotted arrows) and C-EsMV (blue dotted arrows) systems. (b, d) Dynamic analysis under sinusoidal excitation, presenting vibration responses, phase portraits and motion trajectories at (b) $I_{\text{peak}} = 1.4$ A and (d) $I_{\text{peak}} = 2.0$ A.

[1.2. Resonance-induced amplified vibration and the effect of membrane viscoelasticity]

When the input frequency approaches the resonance of the system, even those initially operating in a weakened vibration regime can exhibit amplified vibration (Supplementary Fig. 16 and Fig. 3h). To further examine how the viscoelastic property of the elastic membrane influences resonance and the resulting dynamic behavior, we performed

a parametric analysis (Fig. R3).

At the dynamic viscosity used in our experiments ($\eta = 50 \text{ Pa}\cdot\text{s}$, $\eta/E = 0.001 \text{ s}$), a system that displayed weakened vibration at 5 Hz transitioned to resonance-driven amplified vibration at 30 Hz, with a stable limit cycle maintained (Fig. R3a). However, as the viscosity increased by an order of magnitude, the resonance-induced amplification progressively diminished, and at high damping ($\eta = 50,000 \text{ Pa}\cdot\text{s}$, $\eta/E = 1 \text{ s}$) the resonance effect nearly disappeared. The corresponding discussions have been incorporated into the main text and the Supplementary materials.

Fig. R3. Time-resolved dynamic analysis of membrane motion under resonance at different dynamic viscosity. (a-d) Dynamic responses when the input frequency is shifted from 5 Hz to 30 Hz, showing the effect of resonance gain at various viscosities: (a) $\eta = 50 \text{ Pa}\cdot\text{s}$, (b) $\eta = 500 \text{ Pa}\cdot\text{s}$, (c) $\eta = 5000 \text{ Pa}\cdot\text{s}$, and (d) $\eta = 50,000 \text{ Pa}\cdot\text{s}$.

2. Explanation of vibrational memory effect based on nonlinear dynamics.

[2.1. “Shooting” process and vibrational path using bifurcation diagram]

In Fig. R2a–b, applying $\pm 1.4 \text{ A}$ (attractive/repulsive) was insufficient to cross the barrier between coexisting responses, so the system remained in a low-amplitude state unless resonance was exploited. By contrast, if we intentionally push the system across that barrier—a procedure we call “shooting”—the same bias current (1.4 A) yields amplified vibration that can even persist afterward.

In electrical shooting (Fig. R4a), a brief high-current pulse ($I_{\text{peak}} = 2.0 \text{ A}$) drives the state across the basin boundary. Comparing the bifurcation diagrams before and after shooting at $I_{\text{peak}} = 1.4 \text{ A}$ (Fig. R4b, c), the response follows a

different path on the stable branch, resulting in a larger limit-cycle oscillation without relying on resonance. The persistence of this amplified state upon returning to the lower bias is due to inertia (residual kinetic energy), which sustains oscillation around the new equilibrium branch. This inertia-driven retention represents the vibrational hysteresis-based memory observed in our system. This dynamic hysteresis originates from the inherent hysteretic loop of the C-EsMV system itself, which arises from the static balance between magnetic and elastic forces. The dynamic origin of this vibrational hysteresis is further elaborated in the response to Reviewer 2 (Fig. R9, (Newly added Supplementary Fig. 23)). Similarly, in mechanical shooting (amplified by a transient mechanical perturbation; Fig. R4d), phase-portrait analysis also shows convergence to the amplified limit cycle (bold lines).

Fig.R4. Shooting-enabled branch switching and sustained oscillation. (a) Electrical shooting with a 2.0 A pulse at a 1.4 A bias produces an amplified limit cycle (bold trajectory) (b-c) Bifurcation path at $I = 1.4$ A (b) before and (c) after shooting. (d) Mechanical shooting process and its phase portrait. $f_i = 5$ Hz.

While our experiments emphasize mechanical-triggered memory at a fixed input current, the “shooting” step is not restricted to a specific stimulus. Any perturbation that briefly induces mechanical deformation—**electrical, magnetic, thermal, or pneumatic, delivered with or without contact**—can trigger it. This versatility enables straightforward integration with diverse soft-actuation mechanisms.

[2.2. Analysis of the mechanical memory system and controllability of trigger sensitivity]

Fig. R5. Experimental dynamic vibration response of the mechanical memory in standby and memorized states. (a) Conceptual illustration of the mechanical memory. (b) Time-dependent vibration response and corresponding phase portrait at 5 Hz, comparing the standby (low-amplitude) and memorized (amplified) states. (c) Dynamic vibration response and phase portrait near resonance (30 Hz).

The mechanical memory shows two operating modes: a standby state with weakened vibration and a memorized state with amplified vibration. We analyzed the vibration in each state (Fig. R5). As the data indicate, the memory arises from multistability: two coexisting, dynamically stable limit cycles exist under the same periodic input, and

small disturbances can switch the system between them.

Importantly, our goal is not only to demonstrate its multistability, but also to quantify and control the perturbation required for switching. By adjusting either the peak input current (I_{peak}) or the membrane modulus (E), the trigger force (F_T) needed to change states can be tuned from **0.17 N** up to **63.9 N** (Fig. R6b and c). This threshold tuning provides direct control over sensitivity, which is relevant for engineering applications. In the revised manuscript, we have clarified this interpretation using standard nonlinear dynamics terminology.

Fig. R6. Tunable trigger sensitivity of the mechanical memory. (a) Test configuration under an external trigger force (F_T). (b) Control of F_T by varying the peak input current (I_{peak}). (c) Control of F_T by varying the spacer modulus (E).

[2.3. Effect of membrane viscoelasticity on memory retention]

We examined whether the membrane's viscoelasticity affects memory performance. Increasing the damping coefficient (c) enhances the dissipation of kinetic energy, which progressively suppresses and eventually extinguishes the amplified oscillation. Consequently, the system loses its ability to remain in the memorized state, consistent with standard nonlinear-dynamics expectations that higher damping hinders inertia-assisted switching and reduces the basin of attraction for the high-amplitude limit cycle (Fig. R7).

Fig. R7. Effect of membrane damping on mechanical memory. (a) $\eta = 50$ (b) $\eta = 500$ (c) $\eta = 5000$ (d) $\eta = 50000$ Pa-s.

In summary, by reframing the observed behaviors within the classical framework of nonlinear dynamics, we have clarified that the phenomena described as “memory” and “triggered transitions” originate from multistability, bifurcation, and inertia-driven switching in elasto-magnetic actuators. We fully acknowledge that these are well-established principles in nonlinear oscillation theory; the novelty of our work lies not in redefining them, but in harnessing their interplay within a tunable elasto-magnetic architecture. By treating inertia and magnetic coupling as controllable design parameters, we realize precisely tunable thresholds for state switching, transforming fundamental nonlinear dynamics into a programmable, soft-actuation platform. We are sincerely grateful to the reviewer for prompting this perspective, which allowed us to articulate the system’s true novelty and broader relevance.

Modification/Addition

1. Addition of a supplementary figure and related context describing the bifurcation analysis in both NC-EsMV and C-EsMV systems.

- In page 18 of Supplementary materials

Supplementary Fig. 3. Static analysis of equilibrium and energy as current is varied.

(a) Schematic of the NC-EsMV with variable current (I). (b) State diagram showing equilibrium positions of the NC-EsMV system as a function of input current. (c) Energy landscape of the NC-EsMV system plotted as a function of position for different input currents (-2 , 0 , and 2 A). (d) Schematic of the C-EsMV. (e) State diagram showing equilibrium positions of the C-EsMV system as a function of input current. (f) Energy landscape of the C-EsMV system plotted as a function of position for different input currents (-2 , 0 , and 2 A) (Point A: Saddle-node fold, Point B: Subcritical Hopf).

- In page 3, line 79.

- To analyze the basic behaviors of both systems, we calculated the elasto-magnetic potential energy and the corresponding equilibrium positions as a function of input current. As shown in Supplementary Fig. 3, the NC-EsMV system remains monostable over the entire current range (-2.0 A to $+2.0$ A), and the equilibrium position changes only slightly with current ($\Delta z/D \approx 0.25$). In contrast, the C-EsMV system exhibits bifurcation in behavior, so that the energy landscape alternates between monostable and bistable regimes as the current varies, leading to a much larger shift in equilibrium position ($\Delta z/D > 1$). The appearance of a bistable regime indicates the presence of an elasto-magnetic instability (EsMI), arising from the magnetic attraction between the paired magnets and the elastic restoring force of the membranes.

2. Modification of main Figure 2 for simplification and inclusion of vibration analyses based on both static (equilibrium) and dynamic (time-dependent) behaviors.

1) Original version

2) Revised version

Fig 2. Mechanism of an amplification in EsMV systems.

...

(e – h) Static and dynamic vibration analyses of NC-EsMV and C-EsMV systems. (e) Motion trajectories of each system derived from bifurcation diagram analysis. (f) Time-dependent vibration amplitudes showing dynamic responses of each system. (g) Phase portraits illustrating distinct dynamic states. (h) Dynamic vibration trajectories under sinusoidal input.

- In page 6, line 156.

- Based on the above energy landscape analysis, we further examined the equilibrium transition paths of both systems. In the NC-EsMV, the vibration follows a single monostable branch, with small positional shifts corresponding to minor

equilibrium changes as the current varies between attractive and repulsive phases. In contrast, the C-EsMV exhibits bifurcation in behavior: under increasing attractive current, the magnet moves along the upper stable branch until the potential barrier diminishes and upper state loses stability, leading to an abrupt transition to the lower branch (collapse), as the current reverses, it jumps back—forming a complete hysteretic cycle (Fig. 2e). When dynamic effects are taken into account, the actual displacement (z_{dynamic}) becomes significantly larger than the static equilibrium shift, as shown in Fig. 2f. The corresponding phase portrait confirms that C-EsMV reaches a much higher velocity and converges to a large-amplitude limit cycle compared with NC-EsMV (Fig. 2g). Finally, Fig. 2h illustrates the real-time vibration trajectory under a sinusoidal input, visualizing a single current sweep and the resulting path-dependent oscillation.

3. Addition of a supplementary figure and corresponding discussion on the resonance characteristics and viscoelastic properties of the membrane.

- In page 10, line 249.

It is worth noting that this resonance-induced amplification strongly depends on the viscoelasticity of the membrane. As shown in Supplementary Fig. 17, at the dynamic viscosity used in our experiments ($\eta = 50 \text{ Pa}\cdot\text{s}$, $\eta/E = 0.001 \text{ s}$), a system that exhibited weakened vibration at 5 Hz transitioned to a stable, resonance-driven amplified vibration at 30 Hz. When the dynamic viscosity was increased by one or more orders of magnitude, the amplification effect progressively diminished, and at high damping ($\eta = 50,000 \text{ Pa}\cdot\text{s}$, $\eta/E = 1 \text{ s}$) the resonance response nearly disappeared.

Supplementary Fig. 17. Time-resolved dynamic analysis of membrane motion under resonance at different dynamic viscosities. (a-d) Dynamic responses when the input frequency is shifted from 5 Hz to 30 Hz, showing the effect of resonance gain at various viscosities: (a) $\eta = 50 \text{ Pa}\cdot\text{s}$, (b) $\eta = 500 \text{ Pa}\cdot\text{s}$, (c) $\eta = 5000 \text{ Pa}\cdot\text{s}$, and (d) $\eta = 50,000 \text{ Pa}\cdot\text{s}$.

4. Addition of a supplementary figure and related explanation detailing the “shooting” process and memory retention behavior.

- In page 28 of Supplementary materials

Supplementary Fig. 13. Numerical results of electrical shooting behavior and motion hysteresis in C-EsMV.

(a) Vibration motion of NC-EsMV and C-EsMV at a driving frequency of 5 Hz ($D = 4.0$ mm). Even after the current is increased from 1.4 A to 2.0 A and then returned to 1.4 A, the amplified vibration of C-EsMV is maintained. The shooting process enables C-EsMV to reach the amplified mode at lower currents that were previously unattainable, as clearly seen in the phase portraits. (b–d) Static and dynamic vibration paths at $I_{\text{peak}} = 1.4$ A before shooting: (b) Bifurcation diagram of C-EsMV showing equilibrium positions, (c) corresponding phase portraits of C-EsMV and NC-EsMV, and (d) dynamic vibration trajectory under sinusoidal electrical input, which closely follows the equilibrium path. (e–g) Static and dynamic vibration paths at $I_{\text{peak}} = 1.4$ A after shooting: (e) Bifurcation diagram of C-EsMV showing a switched branch after shooting, (f) phase portraits of C-EsMV and NC-EsMV, and (g) dynamic vibration trajectory under repeated sinusoidal input, exhibiting significantly larger oscillations

despite the same current amplitude.

- In page 9, line 221.

This triggering process, which we term “shooting,” refers to a momentary stimulus that propels the magnet across the energy barrier separating the coexisting states (*weakened* and *amplified*). As shown in Supplementary Fig. 12, when the input current was increased above the onset threshold ($I_{th,on}$) and then reduced (shooting), the system exhibited a sudden transition from a weakened to an amplified vibration state, maintaining the large amplitude even after the current returned to the initial current level ($I_{peak} \approx 1.3$ A). In contrast, systems that never exceeded $I_{th,on}$ (pre-shooting) remained confined to low-amplitude oscillations because the available energy was insufficient to overcome the potential barrier (i.e., the basin boundary between the two attractors). Experimentally, a brief electrical trigger induced this dynamic shift, confirming that the vibration state could be switched and retained through a transient perturbation.

This behavior was further validated through simulation (Supplementary Fig. 13). In the model, a short high-current pulse ($I_{peak} = 2.0$ A) served as the electrical trigger, driving the system across the potential barrier. Comparing the bifurcation diagrams before and after shooting at $I_{peak} = 1.4$ A revealed that the response followed a distinct path along the upper stable branch, resulting in a larger limit-cycle oscillation even in the absence of resonance. The persistence of this amplified vibration after returning to the lower input current arises from inertia, which provides residual kinetic energy to sustain oscillations around the new equilibrium branch. This inertia-driven retention is stably maintained and was further validated as an efficient energy conversion mechanism (Supplementary Fig. 14). It constitutes the vibrational hysteresis-based mechanical memory, which is further analyzed in Fig. 5.

- In page 37 of Supplementary materials.

Supplementary Fig. 22. Numerical verification of mechanically triggered amplification.

Vibration motion of NC-EsMV and C-EsMV systems at an electromagnet-to-magnet distance of (a) $D = 4.0$ mm and (b) $D = 4.5$ mm, both under an input current of 2.0 A. (c) Analogous to the electrical shooting observed in Supplementary Fig. 13, mechanical triggering ($D = 4.5$ mm \rightarrow 4.0 mm \rightarrow 4.5 mm) also sustains amplified vibration in C-EsMV after the trigger is removed, with constant input current. (d) Phase portraits of each system. The C-EsMV exhibits a larger-amplitude limit cycle (bold line) at the same input current and D , indicating greater vibration velocity and displacement compared with NC-EsMV.

- In page 38 of Supplementary materials

< Pre-shooting in mechanical memory >

< Post-shooting in mechanical memory >

Supplementary Fig. 23. Mechanism of vibrational hysteretic behavior in mechanical memory.

Under identical input current I_{peak} , inertia allows two distinct vibration responses: (a) Pre-shooting (weakened state) and (b) post-shooting (amplified state). Without shooting, the magnet oscillates with small displacement near z_{off} (positions 1 – 3) unable to overcome the potential barrier imposed by magnetic attraction. As a result, the system remains in a low-amplitude vibration regime (Point A). By contrast, a brief perturbation (shooting)—which can be electrical, magnetic, or mechanical—pushes the magnet across the barrier to the bottom surface ($z = 0$, position “2”), storing additional elastic energy in the membrane. During the subsequent repulsive phase, this stored energy is released as a large overshoot (“3 → 4”), and the magnet returns with sufficient inertia to repeatedly cross the barrier (“4 → 1 → 5 (= 2)”). The residual kinetic energy from each cycle sustains these large-amplitude oscillations even after the external trigger is removed, corresponding to Point B on the amplitude– I_{peak} curve. If this residual energy is intentionally dissipated (see Supplementary Video 7), the system reverts to the weakened state. Potential energy-position graphs are based on Supplementary Fig. 9e.

- In page 39 of Supplementary materials

Supplementary Fig. 24. Experimental dynamic vibration response of the mechanical memory in standby and memorized states.

(a) Conceptual illustration of the mechanical memory. (b) Time-dependent vibration response and corresponding phase portrait at 5 Hz, comparing the standby (low-amplitude) and memorized (amplified) states. (c) Dynamic vibration response and phase portrait near resonance (30 Hz).

- In page 12, line 309.

This process is analogous to the electrical shooting described earlier, relying on inertia–elastic energy exchange that drives the transition from the weakened to the amplified vibration state (Supplementary Fig. 23). Once triggered, the system sustains oscillation within the amplified branch even after the external input is removed. As shown in Supplementary Fig. 24, this design effectively stores external mechanical stimuli as persistent amplified vibrational states—a phenomenon we term mechanical memory.

- In page 42 of Supplementary materials

Supplementary Fig. 27. Effect of membrane damping on mechanical memory.

(a) $\eta = 50$ (b) $\eta = 500$ (c) $\eta = 5000$ (d) $\eta = 50000$ Pa·s.

- In page 13, line 332.

However, the duration of this non-volatile retention is governed by the viscoelastic property of the membrane (Supplementary Fig. 27): higher viscosity increases damping and gradually suppresses sustained oscillations, whereas a more elastic membrane maintains stable amplified vibration. This confirms that membrane elasticity is essential for robust memory retention.

Comment: 3. To add a little further to the above point, the authors do make general references to the role of inertia in some of the behaviours shown and state that ‘Despite the inherently dynamic nature of instabilities, most existing systems assume quasi-static behaviors, overlooking the fundamental role of inertia during fast state transitions.’ However, structural dynamics explicitly includes inertia in equations of motion as a matter of course, and has been applied to substantially nonlinear systems including those with instability and multistability.

Response: We thank the reviewer for this valuable clarification. We fully agree that inertia is inherently included in structural dynamics and has long been analyzed in nonlinear and unstable systems. Our previous phrasing may have implied otherwise, and we appreciate the opportunity to correct this.

In our system, hysteresis and memory arise only when residual kinetic energy remains after a transition between stable equilibria. This energy produces an overshoot that sustains the hysteretic branch. Thus, inertia is decisive not merely as static mass but as the factor governing the surplus kinetic energy carried through state transitions. As shown in Fig. R7, increasing the dynamic viscosity suppresses this overshoot, leading to the collapse of hysteresis and loss of memory.

To prevent misunderstanding, we have refined the wording in the revised manuscript and added citations to prior nonlinear-dynamics studies that explicitly include inertia. Our intent is not to claim that inertia was previously overlooked, but to demonstrate how its influence on overshoot can be quantitatively tuned and utilized as a design parameter for actuation and memory in soft elasto-magnetic systems.

Modification/Addition

1. Modification of the Introduction to clarify the role of inertia and highlight its significance when coupled with instability-based systems.

- In page 2, line 42.

- Despite the inherently dynamic nature of instabilities, most existing systems assume quasi-static behaviors, overlooking the fundamental role of inertia during fast state transitions. Although some studies have explored the dynamics of instability-driven systems^{31,32}, inertia—defined as the resistance to changes in motion—has generally been regarded as a secondary or disruptive factor rather than central design element. In specific contexts, inertia has been recognized for amplifying motions^{33,34}, yet its broader potential remains underutilized in soft actuation. Harnessing inertial dynamics offers a compelling and largely untapped opportunity to enhance the performance and adaptability of instability-based actuators.

→ Instability-based mechanisms inherently involve rapid, large transitions between distinct states. During these transitions, inertia—the tendency of a body to maintain its motion—is a central element of the dynamics. In structural dynamics^{32,33} and in the broader literature on nonlinear phenomena^{34,35}, inertia governs transient responses and subsequent resettling. In soft actuation, however, it has often been treated as a passive consequence rather than an explicit design parameter. Although prior studies have shown that inertia can amplify motion^{36,37}, its deliberate use to drive transitions between coexisting states or to sustain motion is comparatively less explored. Harnessing both inertia and mechanical instabilities therefore offers a practical design strategy for soft actuators with improved adaptability and energy efficiency.

2. Addition of references related to nonlinear dynamics and its applications.

- In page 19 (References).

34. Habib, G. Predicting saddle-node bifurcations using transient dynamics: a model-free approach. *Nonlinear Dyn.* **111**, 20579–20596 (2023)

35. Rega, G. Nonlinear dynamics in mechanics: state of the art and expected future developments. *J. Comput. Nonlinear Dyn.* **17**, 080802 (2022).

Comment: 4. I also think some of the claims for energy efficiency and ‘amplifying kinetic energy’ need to be stated with a great degree of care - physics tells us that energy is never created, however an oscillating system can contain a large amount of conserved energy that it exchanges between kinetic and potential energy during oscillation, and these oscillations can therefore be sustained with very little ongoing energy input. So to imply that the system is giving us any energy for free is simply wrong, however I think examples such as striking the suspended ball do reveal that exploiting instabilities in these ways can allow energy to be released at a very high rate, ie. it is accurate to describe these systems as potentially showing very high power over short periods.

Response: We sincerely thank the reviewer for this thoughtful and detailed comment. We also agree that our description must not imply any violation of energy conservation. Our study does not claim spontaneous energy generation; rather, the term “efficiency” refers to the conversion efficiency—that is, how effectively the externally supplied electrical energy to the electromagnet is transformed into the kinetic energy of the vibrating magnets.

In the conventional NC-EsMV system, only a small fraction of the supplied electromagnetic energy contributes to motion; most of the field energy is dissipated to the surroundings. In contrast, the C-EsMV configuration enhances magnetic coupling, allowing a greater portion of the input energy to be temporarily stored as elastic energy in the membrane and then released as kinetic energy during collapse–release events. This process results in a measured conversion efficiency over 700-fold higher than that of the NC-EsMV, as detailed in the Supplementary Note 6.

The phrase “amplified kinetic energy” was originally intended to convey this improved energy conversion and

transient release rate, not an actual amplification or creation of energy. Nevertheless, we fully recognize that the wording could be misleading. To avoid ambiguity, we have revised the manuscript to use more precise expressions such as “conversion efficiency,” “motion amplification,” or “enhanced utilization of input energy.” We have also added a clear definition of conversion efficiency and its calculation method in the main text. We thank the reviewer for highlighting this important point, which has allowed us to improve both the accuracy and clarity of our presentation.

Modification/Addition

1. Modification of “energy efficiency” to clarify it as “energy conversion efficiency.”

1) In page 2, line 53.

~ capable of amplifying kinetic energy by over three orders of magnitude compared to a non-coupled control system (NC-EsMV) (Fig. 1a and b).

→ ~ capable of **enhancing kinetic energy conversion** by over three orders of magnitude compared to a non-coupled control system (NC-EsMV) (Fig. 1a and b).

2) In page 6, line 152. (original content moved to Supplementary Fig. 5)

This stands in sharp contrast to the C-EsMV system, which undergoes four distinct stages in each vibrational cycle: i) loading, ii) take-off, iii) vibration, and iv) landing, marked by vertical dashed lines. Initially, the magnet collapses onto the electromagnet’s surface, storing maximum elastic potential energy (i, $0 \text{ s} < t < 0.077 \text{ s}$). When electromagnetic repulsion increases sufficiently to yield a net positive force, it triggers the take-off (ii, $t = 0.077 \text{ s}$). After take-off, the system enters a vibration phase governed by its natural frequency (iii, $0.102 \text{ s} < t < 0.132 \text{ s}$), then returns to the electromagnet’s surface to complete the cycle (iv, $0.132 \text{ s} \leq t \leq 0.2 \text{ s}$). In the C-EsMV system, magnetic attraction establishes a critical threshold that facilitates significant elastic to kinetic energy exchange.

→ **In contrast, the C-EsMV system establishes a critical threshold that enables efficient conversion of electrical input into mechanical motion. This configuration enables improved conversion efficiency through elasto–magnetic coupling, allowing a larger portion of the supplied power to be stored and released as kinetic energy.**

3) In page 9, line 210.

we analyzed the ratio of the maximum kinetic energy of C-EsMV to NC-EsMV ($\eta_{C/NC}$) as a function of I_{peak} with $D = 4.0$ mm.

→ we analyzed the ratio of energy conversion efficiencies ($\epsilon_{C/NC}$, the ratio of the maximum kinetic energy of C-EsMV to that of NC-EsMV under identical electrical input) as a function of I_{peak} with $D = 4.0$ mm.

4) In page 10, line 256.

System efficiency can be enhanced by waveform design (Supplementary Note 5, Fig. 3i), as the waveform shape directly affects the input energy required for amplified vibrations.

→ The energy conversion efficiency of the system—defined as the ratio of the maximum kinetic energy of the vibrating magnet to the supplied electrical energy—is relatively low (< 1 %) because of the limited coil turns and associated geometric factors of the electromagnet. However, this efficiency can be substantially enhanced by waveform design (Supplementary Note 6, Fig. 3i), as the waveform shape directly affects the input energy required for amplified vibrations.

2. Addition of a statement describing high-rate energy release and short-term power output.

- In page 11, line 281.

~, with the force rising from 68 mN at $I_{peak} = 0.8$ A to over 225 mN at $I_{peak} = 0.9$ A (1 Hz, Fig. 4d).

→ ~, with the force rising from 68 mN at $I_{peak} = 0.8$ A to over 225 mN at $I_{peak} = 0.9$ A (1 Hz, Fig. 4d), releasing stored elastic energy at a high rate over a short period, resulting in high power output.

Comment: 5. Despite this negative review I do encourage the authors to continue the work and am sure that many publications will ensue. The devices certainly capture the imagination, in particular for the suggested applications of a flutter actuator and a ‘mechanical transistor’.

Response: We sincerely appreciate the reviewer’s encouraging remarks regarding the originality and potential of our

work. We are grateful for the recognition of its broader applicability, including prospective uses as flutter actuators and mechanical-transistor-like devices. We will continue to explore these directions and further develop the concept, and we hope that the present study provides a foundation for future advances in this field.

Reviewer #2

Remarks to the Author: This manuscript reports on elasto-magnetic instabilities that can enable gigantic actuations and mechanical memory. The authors have designed a coupled elasto-magnetic vibration system with a pair of magnets and elastic membrane. Both the actuated force and displacement are much higher than a non-coupled control system. The work is original in working principle and the results of mechanical memory are impressive. The data shown in main text and extended data can support the conclusions. Thus, I recommend the publication of this manuscript after minor revision. The following issues need to be addressed by the authors:

Response: We sincerely thank the reviewer for the positive and encouraging assessment of our work. We deeply appreciate your recognition of the originality of the elasto-magnetic system and the concept of mechanical memory. We have carefully addressed all comments, and revised manuscript accordingly to improve clarity of explanations and to better highlight the physical mechanisms and practical implications of our results. Detailed responses to each point are provided below.

Comment: 1. In the manuscript, the distance between the magnet and coil D is carefully investigated as a parameter. In the design of the coupling system, will the size of the magnet and the elastic membrane affect the vibrational displacement? What is the intensity of magnetization for the magnet?

Response: We thank the reviewer for this insightful question regarding the influence of geometric parameters and magnetization strength. As noted, in the coupled elasto-magnetic vibration (C-EsMV) system, both the initial magnet–coil distance (D) and the dimensions of the permanent magnet and elastic membrane are key parameters that determine the dynamic behavior.

Revisiting Newton's second law, the governing equation of motion can be written as:

$$m\ddot{z} = F_{ES} - F_{MM} - F_{EM} - F_d,$$

where each force term explicitly depends on the magnet and membrane design parameters. Although our main text focused on the effect of D , we agree that the magnet size and modulus of membrane also strongly affect the vibrational

displacement.

To generalize the design beyond a single geometry, we introduced two characteristic forces, the elastic restoring force when the membrane achieves a radial stretch $\lambda_r = \sqrt{2}$ at the inclination angle of 45° (F_{ES}^*) and the magnet-magnet attraction force at $z = 0$ (F_{MM}^*) (as detailed in Supplementary Note. 1):

$$F_{ES}^* = \frac{\pi E}{2(1+\nu)} h_m \lambda_p^2 (R_m + R_a),$$

$$F_{MM}^* = \frac{\mu_0}{2} M_a M_b \Omega|_{z=0} =$$

$$\frac{\mu_0}{2} M_a M_b R_a R_b \int_0^{2\pi} \cos \delta d\delta \int_{-\frac{h_a}{2}}^{\frac{h_a}{2}} \left[\frac{1}{\sqrt{R_a^2 + R_b^2 - 2R_a R_b \cos \delta + \left(\frac{h_a}{2} + h - z_a\right)^2}} - \frac{1}{\sqrt{R_a^2 + R_b^2 - 2R_a R_b \cos \delta + \left(\frac{h_a}{2} + h + h_b - z_a\right)^2}} \right] dz_a$$

When these forces are balanced ($F_{ES}^* \approx F_{MM}^*$), the C-EsMV system consistently exhibits similar amplified vibration and mechanical memory across different scales. For example, by varying the membrane modulus (E), pre-stretch ratio (λ), and thickness (h), we observed comparable amplification even for different geometries (Supplementary Fig. 11).

Supplementary Fig. 11. Initial position–amplitude plots of the plot of C-EsMV system with varying conditions of elastic membrane, but with identical characteristic membrane force F_{ES}^* .

If this balance is broken—when magnetic force dominates, the system collapses; when elastic restoring force dominates, it remains in a weakened, low-amplitude state. As shown in Fig. R8a, both magnet and membrane geometries shift these regimes by changing the characteristic forces. Once F_{ES}^* and F_{MM}^* are fixed, the vibration state can be further tuned by adjusting D or the peak input current (I_{peak}) (Fig. R8b, c).

Fig. R8. Schematic illustration of parameter control for tuning the C-EsMV system.

Regarding magnetic properties, the permanent magnets used (NdFeB, radius = 4 mm, thickness = 0.5 mm, three-layer stack) have a typical magnetization of $M_a = M_b = 4 \times 10^5$ A/m, as confirmed experimentally by measuring the attraction force as a function of distance. This value was consistently used in both experiments and simulations.

Modification/Addition

1. Addition of information on the magnetization of permanent magnets used in both experiments and simulations.

- In page 16, line 375 (Method section).

Both NC-EsMV and C-EsMV systems utilized identical NdFeB permanent magnets (Jungsin Corp.) and electromagnets (TDK Corp., WR151580-48F2-G). The permanent magnets (radius = 4 mm, thickness = 0.5 mm, three-layer stack) exhibit a typical magnetization of $M_a = M_b = 4 \times 10^5$ A/m, which was experimentally verified by measuring the magnetic attraction force as a function of distance. This magnetization value was consistently used in both experiments and simulations.

Comment: 2. Why do both systems exhibit a rapid decay in vibration amplitude beyond resonance (Line 104)? Is there any high-order resonance frequency for the system?

Response: We thank the reviewer for this insightful question. Experimentally, both NC-EsMV and C-EsMV exhibit a noticeable decay in vibration amplitude once the driving frequency exceeds resonance. At resonance, the input energy is efficiently accumulated in phase with the magnet motion; however, beyond resonance, the increasing phase mismatch between the motion and the input signal inhibits effective energy conversion, resulting in eventual decay of the vibration amplitude.

Unlike NC-EsMV, where the amplitude rises sharply and symmetrically around resonance, the C-EsMV shows a distinctly asymmetric response. At lower frequencies, the system surpasses the threshold to generate amplified vibration, and as the frequency approaches resonance, this effect further enhances the vibration amplitude. Beyond resonance, however, the increasing phase mismatch reduces effective acceleration, preventing the system from overcoming the threshold and causing a sudden transition to weakened vibration. To avoid ambiguity, the term “rapid decay” in the original text has been replaced with “eventual decay,” which more accurately describes this gradual loss of amplitude following the abrupt state transition.

Additionally, in the original manuscript, we referred to the resonance frequency as the point at which the vibration amplitude reaches its maximum under AC excitation. In reality, this corresponds to the nonlinear forced resonance of the coupled magnet–membrane system rather than the intrinsic free-vibration frequency.

To quantitatively describe this behavior, the intrinsic baseline frequency of the magnet–membrane pair, in the absence of magnetic and damping forces, can be given by

$$\omega_0 = \sqrt{\frac{k_{ES}}{m}},$$

where k_{ES} is the elastic membrane spring constant and m is the magnet–membrane mass. When magnetic and damping forces are included, the damped frequency becomes

$$\omega_d = \sqrt{\frac{k_{ef}(z^*)}{m} - \left(\frac{c_{ef}}{2m}\right)^2},$$

where c_{ef} denotes the damping coefficient of the membrane, and $k(z^*)$ varies with the equilibrium position z^* .

Therefore, the observed resonance in our experiments corresponds to a nonlinear forced response whose frequency shifts dynamically with the equilibrium position and the applied current, rather than a single fixed natural frequency.

Regarding higher-order resonances: in principle, the membrane, as a continuous elastic structure, supports multiple natural modes. In our experiments, however, the excitation was applied near the fundamental mode, and phase mismatch and potential damping effect of membrane strongly suppresses higher-order responses. As a result, the dynamics we observed were dominated by the fundamental resonance. We have clarified these points in the revised manuscript and noted that multimode or subharmonic responses could emerge under alternative geometries or excitation conditions.

Modification/Addition

1. Addition of an explanation of the resonance frequency in our system.

- In page 9 of Supplementary materials.

Supplementary Note 3. Resonant frequency in our system

In our system, the observed resonance corresponds to the nonlinear forced resonance of the coupled magnet–membrane system rather than the intrinsic free-vibration frequency. Consequently, the response around resonance is asymmetric: the vibration amplitude increases gradually as the frequency approaches resonance, but exhibits an eventual decay beyond the resonance frequency. At lower frequencies, the system surpasses the threshold to generate amplified vibration, and as the frequency approaches resonance, this effect further enhances the amplitude. Beyond resonance, however, the increasing phase mismatch between the magnet motion and the input signal reduces effective acceleration, preventing the system from overcoming the threshold and leading to weakened vibration and eventual decay.

This asymmetry primarily arises from the phase mismatch–induced loss of energy transfer efficiency at higher frequencies. In addition, the strongly nonlinear magnetic force, which intensifies as the inter-magnet distance decreases, further enhances this asymmetric response. Together, these effects cause the frequency response to deviate from the symmetric Lorentz-type profile typical of linear oscillators.

The intrinsic baseline frequency of the magnet–membrane pair, in the absence of magnetic and damping forces, can be given by the mass–spring relation:

$$\omega_0 = \sqrt{\frac{k_{ES}}{m}} \quad (9)$$

where k_{ES} is the spring constant of membrane and m is the magnet–membrane mass.

When magnetic and damping forces are included, the system can be linearized locally around an equilibrium position z^* :

$$m z'' + c_{ef} z' + k_{ef} z = 0, \quad (10)$$

$$k_{ef} = \frac{\delta(F_{ES}-F_{MM})}{\delta z} \Big|_{z=z^*},$$

The corresponding damped frequency is given by:

$$\omega_d = \sqrt{\frac{k_{ef}(z^*)}{m} - \left(\frac{c_{ef}}{2m}\right)^2}, \quad (11)$$

where c_{eff} denotes the effective damping coefficient of the membrane, which in our system may depend not only on velocity but also on displacement due to the viscoelastic membrane behavior and contact conditions. $k_{eff}(z^*)$ varies dynamically with the equilibrium position determined by the applied magnetic forces. Thus, the resonant frequency shifts continuously with both equilibrium position and input current, confirming that the observed resonance arises from a nonlinear forced response rather than a fixed natural frequency.

For small oscillations around an equilibrium z^* , $c(z)$ can be approximated as constant, yielding a well-defined local damped natural frequency. However, in the amplified state, the C-EsMV traverses both equilibria with large excursions, where the restoring and damping forces vary significantly within each cycle. The motion is therefore better described as a nonlinear forced periodic response (limit cycle) governed by the AC electromagnetic drive, whose magnitude depends on displacement while its temporal variation follows the input frequency. This displacement dependence introduces nonlinear stiffness effects, broadening and shifting the resonance relative to eq. (9).

Although the membrane can, in principle, support multiple vibration modes, our experiments operated near the fundamental resonance, where phase mismatch and potential damping effect of membrane strongly suppresses higher-order responses. Weak secondary resonances may appear under certain conditions but are highly sensitive to asymmetry and nonlinear coupling. If geometric or material asymmetry were introduced, multimode or non-axisymmetric oscillations could emerge, leading to richer nonlinear dynamics under modified designs or excitation conditions.

Comment: 3. The demo of mechanical memory in Fig. 5 is interesting. More description for the working mechanism is needed. The guess is that the hysteric behavior in the plot of Amplitude-Ipeak is key to the memory behavior. Why is there a hysteresis in this curve?

Response: We thank the reviewer for this thoughtful inquiry into the origin of the hysteresis underlying the memory behavior. Indeed, the amplitude– I_{peak} curve is the key to understanding the memory demonstration in Fig. 5. The hysteresis allows the system, at the same current, to remain in either a weakened vibration state or to switch into an amplified state when triggered.

The origin of this vibrational hysteresis lies in inertia—the residual kinetic energy retained during actuation. As shown in Fig. R9, we compared the system behavior with and without the shooting process, in which the magnet is momentarily driven across the energy barrier separating the two stable states (branches) of the bistable energy landscape.

Without shooting, even at the same I_{peak} , the system remains in the low-amplitude vibration regime (Point A in Fig. R9a,b). Mechanistically, when in this weakened state, the magnet oscillates with small displacement near z_{off} , unable to overcome the potential barrier imposed by magnetic attraction. Consequently, motion is confined between positions “1” and “3.”

< Pre-shooting in mechanical memory >

< Post-shooting in mechanical memory >

Fig. R9. Mechanism of hysteretic behavior in mechanical memory. Under identical input I_{peak} , inertia enables distinct vibration responses: (a) without shooting (weakened state) and (b) with shooting (amplified state).

By contrast, when a brief perturbation (shooting)—electrical, magnetic, or mechanical—is applied (Fig. R9c,d), the magnet is pushed across the barrier toward the bottom surface (position “2,” $z = 0$), storing additional elastic energy in the membrane. During the subsequent repulsive phase, this stored energy is released as a large overshoot, driving the magnet beyond the static equilibrium position (“3” \rightarrow “4”) and then back (“1” \rightarrow “2 = 5”). The residual kinetic energy from each cycle sustains these large-amplitude oscillations even after the external trigger is removed, corresponding to Point B on the amplitude– I_{peak} curve. If this residual energy is intentionally dissipated (see Supplementary Video 7), the system reverts to the weakened state.

As further detailed in the revised manuscript and in the Response to Reviewer 1 (Figs. R1–R3), the coupled elastomagnetic vibration (C-EsMV) system exhibits bistability arising from the interplay between magnetic attraction and

elastic restoring forces. The corresponding bifurcation diagram illustrates how the system transitions between distinct stable branches before and after the shooting event during vibration. In addition, the newly added Supplementary Fig. 23 (Fig. R9) clarifies why the same input current can yield different oscillation amplitudes through branch switching behavior.

Modification/Addition

1. Addition of a Supplementary Figure and corresponding description presenting the bifurcation analysis of both systems, placed before the vibrational memory section.

- In page 18 of Supplementary materials

Supplementary Fig. 3. Static analysis of equilibrium and energy as current is varied.

(a) Schematic of the NC-EsMV with variable current (I). (b) State diagram showing equilibrium positions of the NC-EsMV system as a function of input current. (c) Energy landscape of the NC-EsMV system plotted as a function of position for different input currents (-2 , 0 , and 2 A). (d) Schematic of the C-EsMV. (e) State diagram showing equilibrium positions of the C-EsMV system as a function of input current. (f) Energy landscape of the C-EsMV system plotted as a function of position for different input currents (-2 , 0 , and 2 A) (Point A: Saddle-node fold, Point B: Subcritical Hopf).

- In page 3, line 79.

- To analyze the basic behaviors of both systems, we calculated the elasto-magnetic potential energy and the corresponding

equilibrium positions as a function of input current. As shown in Supplementary Fig. 3, the NC-EsMV system remains monostable over the entire current range (-2.0 A to $+2.0$ A), and the equilibrium position changes only slightly with current ($\Delta z/D \approx 0.25$). In contrast, the C-EsMV system exhibits bifurcation in behavior, so that the energy landscape alternates between monostable and bistable regimes as the current varies, leading to a much larger shift in equilibrium position ($\Delta z/D > 1$). The appearance of a bistable regime indicates the presence of an elasto-magnetic instability (EsMI), arising from the magnetic attraction between the paired magnets and the elastic restoring force of the membranes.

2. Addition of explanations validating and describing the shooting behavior responsible for memorization.

- In page 28 of Supplementary materials

Supplementary Fig. 13. Numerical results of electrical shooting behavior and motion hysteresis in C-EsMV.

(a) Vibration motion of NC-EsMV and C-EsMV at a driving frequency of 5 Hz ($D = 4.0$ mm). Even after the current is increased from 1.4 A to 2.0 A and then returned to 1.4 A, the amplified vibration of C-EsMV is maintained. The shooting process enables C-EsMV to reach the amplified mode at lower currents that were previously unattainable, as clearly seen in the phase portraits. (b–d) Static and dynamic vibration paths at $I_{\text{peak}} = 1.4$ A before shooting: (b) Bifurcation diagram of C-EsMV showing equilibrium positions, (c) corresponding phase portraits of C-EsMV and NC-EsMV, and (d) dynamic vibration trajectory under sinusoidal electrical input, which closely follows the equilibrium path.

(e–g) Static and dynamic vibration paths at $I_{\text{peak}} = 1.4$ A after shooting: (e) Bifurcation diagram of C-EsMV showing a switched branch after shooting, (f) phase portraits of C-EsMV and NC-EsMV, and (g) dynamic vibration trajectory under repeated sinusoidal input, exhibiting significantly larger oscillations despite the same current amplitude.

- In page 9, line 221.

- This triggering process, which we term “shooting,” refers to a momentary stimulus that propels the magnet across the energy barrier separating the coexisting states (*weakened and amplified*). As shown in Supplementary Fig. 12, when the

input current was increased above the onset threshold ($I_{th,on}$) and then reduced (shooting), the system exhibited a sudden transition from a weakened to an amplified vibration state, maintaining the large amplitude even after the current returned to the initial level ($I_{peak} \approx 1.3$ A). In contrast, systems that never exceeded $I_{th,on}$ (pre-shooting) remained confined to low-amplitude oscillations because the available energy was insufficient to overcome the barrier. Experimentally, a brief electrical trigger induced this dynamic shift, confirming that the vibration state could be switched and retained through a transient perturbation.

This behavior was further validated through simulation (Supplementary Fig. 13). In the model, a short high-current pulse ($I_{peak} = 2.0$ A) served as the electrical trigger, driving the system across the potential barrier. Comparing the bifurcation diagrams before and after shooting at $I_{peak} = 1.4$ A revealed that the response followed a distinct path along the upper stable branch, resulting in a larger limit-cycle oscillation even in the absence of resonance. The persistence of this amplified vibration after returning to the lower input current arises from inertia, which provides residual kinetic energy to sustain oscillations around the new equilibrium branch. This inertia-driven retention is stably maintained and was further validated as an efficient energy conversion mechanism (Supplementary Fig. 14). It constitutes the vibrational hysteresis-based mechanical memory, which is further analyzed in Fig. 5.

- In page 37 of Supplementary materials

Supplementary Fig. 22. Numerical verification of mechanically triggered amplification.

Vibration motion of NC-EsMV and C-EsMV systems at an electromagnet-to-magnet distance of (a) $D = 4.0$ mm and (b) $D = 4.5$ mm, both under an input current of 2.0 A. (c) Analogous to the electrical shooting observed in Supplementary Fig. 13, mechanical triggering ($D = 4.5$ mm \rightarrow 4.0 mm \rightarrow 4.5 mm) also sustains amplified vibration in C-EsMV after the trigger is removed, with constant input current. (d) Phase portraits of each system. The C-EsMV exhibits a larger-amplitude limit cycle (bold line) at the same input current and D , indicating greater vibration velocity and displacement compared with NC-EsMV.

3. Addition of a Supplementary Figure and discussion emphasizing the central role of inertia in enabling the memorization effect.

- In page 38 of Supplementary materials

Supplementary Fig. 23. Mechanism of vibrational hysteretic behavior in mechanical memory.

Under identical input current I_{peak} , inertia allows two distinct vibration responses: (a) Pre-shooting (weakened state) and (b) post-shooting (amplified state). Without shooting, the magnet oscillates with small displacement near z_{off} (positions 1 – 3) unable to overcome the potential barrier imposed by magnetic attraction. As a result, the system remains in a low-amplitude vibration regime (Point A). By contrast, a brief perturbation (shooting)—which can be electrical, magnetic, or mechanical—pushes the magnet across the barrier to the bottom surface ($z = 0$, position “2”), storing additional elastic energy in the membrane. During the subsequent repulsive phase, this stored energy is released as a large overshoot (“3 \rightarrow 4”), and the magnet returns with

sufficient inertia to repeatedly cross the barrier (“4 → 1 → 5 (= 2)”). The residual kinetic energy from each cycle sustains these large-amplitude oscillations even after the external trigger is removed, corresponding to Point B on the amplitude– I_{peak} curve. If this residual energy is intentionally dissipated (see Supplementary Video 7), the system reverts to the weakened state. Potential energy-position graphs are based on Supplementary Fig. 9e.

- In page 12, line 309.

This process is analogous to the electrical shooting described earlier, relying on inertia–elastic energy exchange that drives the transition from the weakened to the amplified vibration state (Supplementary Fig. 23). Once triggered, the system sustains oscillation within the amplified branch even after the external input is removed. As shown in Supplementary Fig. 24, this design effectively stores external mechanical stimuli as persistent amplified vibrational states—a phenomenon we term mechanical memory.

Comment: 4. In the R3 regime of mechanical memory, the system is non-volatile. The elastic membrane is vibrating continuously. Will the viscoelasticity of the polymeric membrane affect the vibration and mechanical memory?

Response: We thank the reviewer for this thoughtful comment regarding the role of viscoelasticity in the observed mechanical memory behavior. To address this, we conducted simulations in which the dynamic viscosity (η) was varied, using $\eta = 50 \text{ Pa}\cdot\text{s}$ as the experimental baseline measured from membrane free-vibration tests.

As shown in Fig. R7, starting from a bottom-attached configuration with an additional initial energy input (“shooting”), the system maintained stable amplified oscillations at $\eta = 50 \text{ Pa}\cdot\text{s}$. The corresponding phase-portrait trajectories confirmed convergence to a single stable limit cycle.

Fig. R7. Effect of membrane damping on mechanical memory. (a) $\eta = 50$ (b) $\eta = 500$ (c) $\eta = 5000$ (d) $\eta = 50000$ Pa·s. $f_i = 30$ Hz.

When η was increased by one, two, and three orders of magnitude ($\eta = 500, 5000, 50000$ Pa·s), the amplified motion was progressively suppressed, and the motion eventually decayed to the weakened state. The steady amplitude decreased systematically with increasing η , indicating that viscous dissipation strongly influences not only vibration amplitude but also the stability of the memory state.

In all simulations, the elastic modulus was fixed at 50 kPa (consistent with experiments), so that the range of dynamic viscosity corresponds to characteristic relaxation times $\tau = \eta/E$ from 1 ms to 1 s—covering the transition from elasticity-dominated to viscosity-dominated responses. Although the exact dependence on the dimensionless parameter $\omega\tau$ varies with the viscoelastic model used, the trend is clear: increasing τ leads to stronger dissipation, suppression of amplification, and ultimately elimination of persistent memory.

These findings directly confirm the reviewer’s point that viscous effects are essential and can shift—or even remove—the operational memory regimes of the system.

Modification/Addition

1. Addition of an explanation on how the membrane’s viscoelasticity affects memorization.
 - 1) In page 42 of Supplementary materials

Supplementary Fig. 27. Effect of membrane damping on mechanical memory. (a) $\eta = 50$ (b) $\eta = 500$ (c) $\eta = 5000$ (d) $\eta = 50000$ Pa-s.

1) In page 13, line 332.

However, the duration of this non-volatile retention is governed by the viscoelastic property of the membrane (Supplementary Fig. 27): as discussed earlier, higher viscosity increases damping and gradually suppresses sustained oscillations, whereas a more elastic membrane maintains stable amplified vibration. This confirms that membrane elasticity is essential for robust memory retention.

Reviewer #3

Remarks to the Author: This manuscript presents a very interesting and useful electromagnetically driven actuator system. The authors introduce an Elasto-Magnetic Vibration (C-EsMV) mechanism that leverages inertia-driven dynamics and modulated bistability arising from the interplay between magnetic attraction and elastic tension, thereby amplifying both force output and displacement. Furthermore, the concept of inertial vibration hysteresis is used to demonstrate different forms of mechanical memory with tunable performance. Overall, the system is well designed, the concept is convincing, and the data and analysis are carried out in a systematic manner. This work deserves publication in Nature Communications.

Response: We sincerely thank the reviewer for the positive and encouraging evaluation of our work. We greatly appreciate your recognition of the design, concept, and systematic analysis of the elasto-magnetic vibration (C-EsMV) system. We have carefully addressed all comments and revised the manuscript accordingly to improve the clarity and completeness of our explanations. Detailed responses to each point are provided below.

Comment: 1. However, the manuscript seems to have been structured with Nature in mind, which results in the writing and discussion in the main text being rather condensed. As a consequence, much of the essential data has been placed in the Extended Data and Supplementary Figures, making the overall discussion less easy to follow. In addition, some figures contain a large amount of information but only limited explanation, which makes interpretation less straightforward. I would suggest the authors revise the manuscript by expanding the discussion and improving clarity.

Response: We appreciate the reviewer's thoughtful comments on the overall structure and clarity of the manuscript. We agree that the original version, which followed the condensed Nature format, placed much of the essential data in the Extended Data and Supplementary Figures, which may have made parts of the discussion less straightforward to follow.

In the revised manuscript, we have:

- **Reorganized** and **Simplified** figures, reduced visual density and provided clearer explanations in both the captions and corresponding text.
- **Expanded the mechanistic discussion of the memory behavior**, making the link between experimental results and physical interpretation clearer.

These revisions were applied throughout the Introduction, Results, and Discussion sections to enhance readability and clarity. We believe that the improved structure more effectively conveys the significance and coherence of our findings.

Comment: 2. Understanding of the instability. On line 70, the authors describe: “The resulting combination of magnetic and elastic force generates an elasto-magnetic instability (EsMI) with two stable states (Figure 1d).” Since the electromagnet under AC current can show “off, attract, repulse” states, how should one understand the concept of two stable states in this multi-state context? It seems that the authors use the static equilibrium (off state) as the reference, but the relation to instability needs clearer explanation.

Response: We thank the reviewer for this insightful and important comment. We acknowledge that our original description of the elasto-magnetic instability (EsMI) may have been ambiguous in the context of the “off,” “attract,” and “repulse” states of the electromagnet.

When the electromagnet is **off** (Fig. R10), the static balance between magnetic attraction and elastic restoring force produces a bistable energy landscape with two equilibrium positions: one near the initial membrane–magnet spacing ($z = z_{\text{off}}$) and another where the magnet adheres to the electromagnet surface ($z = 0$).

Fig. R10. Potential energy landscape of the C-EsMV system when the electromagnet is off, showing two stable equilibria (bistable state).

When the electromagnet is driven by an AC current, the alternating magnetic field dynamically reshapes this potential landscape (Fig. R11). At the extrema of maximum attraction or repulsion, only one minimum exists, rendering the system instantaneously monostable. Furthermore, as shown in Fig. R11b, the magnet does not stop precisely at this static equilibrium point; instead, inertia carries it beyond the equilibrium, resulting in an overshoot that initiates the next cycle of motion.

Fig. R11. Potential energy landscapes when the electromagnet is **on**: (a) under maximum attraction and (b) under maximum repulsion. In (b), the magnet overshoots the static equilibrium due to inertia, initiating the next oscillation cycle.

Thus, the coupled EsMV system behaves as a time-dependent nonlinear oscillator that periodically transitions between bistable and monostable configurations within each AC cycle. The observed EsMI arises not from the static potential alone, but from the dynamic modulation of the bistable landscape combined with inertia-driven overshoot.

To further clarify this mechanism, we have added a bifurcation diagram in Supplementary Fig. 3 (= Fig. R1) showing how the system alternates between monostable and bistable states with varying external current. Additionally, Fig. 2 now includes the dynamic vibration trajectories and phase portraits, visualizing how the magnet moves across the potential wells in real time.

These additions make explicit how static bistability, dynamic instability, and inertia jointly contribute to the elastomagnetic vibration behavior.

Modification/Addition

1. Addition of Supplementary figure and related context about bifurcation analysis in both systems.

- In page 18 of Supplementary materials

Supplementary Fig. 3. Static analysis of equilibrium and energy as current is varied.

(a) Schematic of the NC-EsMV with variable current (I). (b) State diagram showing equilibrium positions of the NC-EsMV system as a function of input current. (c) Energy landscape of the NC-EsMV system plotted as a function of position for different input currents (-2 , 0 , and 2 A). (d) Schematic of the C-EsMV. (e) State diagram showing equilibrium positions of the C-EsMV system as a function of input current. (f) Energy landscape of the C-EsMV system plotted as a function of position for different input currents (-2 , 0 , and 2 A) (Point A: Saddle-node fold, Point B: Subcritical Hopf).

- In page 3, line 79.

- To analyze the basic behaviors of both systems, we calculated the elasto-magnetic potential energy and the corresponding equilibrium positions as a function of input current. As shown in Supplementary Fig. 3, the NC-EsMV system remains monostable over the entire current range (-2.0 A to $+2.0$ A), and the equilibrium position changes only slightly with current ($\Delta z/D \approx 0.25$). In contrast, the C-EsMV system exhibits bifurcation in behavior, so that the energy landscape alternates between monostable and bistable regimes as the current varies, leading to a much larger shift in equilibrium position ($\Delta z/D > 1$). The appearance of a bistable regime indicates the presence of an elasto-magnetic instability (EsMI), arising from the magnetic attraction between the paired magnets and the elastic restoring force of the membranes.

2. Modification of main Figure 2 for simplification and inclusion of vibration analyses based on both static (equilibrium) and dynamic (time-dependent) behaviors.

1) Original version (original content moved to Supplementary Fig. 5)

2) Revised version

Fig 2. Mechanism of an amplification in EsMV systems.

...

(e – h) Static and dynamic vibration analyses of NC-EsMV and C-EsMV systems. (e) Motion trajectories of each system derived from bifurcation diagram analysis. (f) Time-dependent vibration amplitudes showing dynamic responses of each system. (g) Phase portraits illustrating distinct dynamic states. (h) Dynamic vibration trajectories under sinusoidal input.

- In page 6, line 156.

- Based on the above energy landscape analysis, we further examined the equilibrium transition paths of both systems. In the NC-EsMV, the vibration follows a single monostable branch, with small positional shifts corresponding to minor equilibrium changes as the current varies between attractive and repulsive phases. In contrast, the C-EsMV exhibits bifurcation in behavior: under increasing attractive current, the magnet moves along the upper stable branch until the potential barrier diminishes and upper state loses stability, leading to an abrupt transition to the lower branch (collapse), as the current reverses, it jumps back—forming a complete hysteretic cycle (Fig. 2e). When dynamic effects are taken into account, the actual displacement ($z_{dynamic}$) becomes significantly larger than the static equilibrium shift, as shown in Fig. 2f. The corresponding phase portrait confirms that C-EsMV reaches a much higher velocity and converges to a large-amplitude limit cycle compared with NC-EsMV (Fig. 2g). Finally, Fig. 2h illustrates the real-time vibration trajectory under a sinusoidal input, visualizing a single current sweep and the resulting path-dependent oscillation.

Comment: 3. The stepwise response and large-amplitude, high-efficiency actuation of C-EsMV compared to NC-EsMV are clearly shown in Figure 1e,f. However, if the input current is further increased, will the maximum amplitude of NC-EsMV eventually approach that of C-EsMV, since the contribution of the coupled magnets is no longer the dominant factor in the overall magnetic force? This is consistent with Figure 1f, in which the amplitude difference between C-EsMV and NC-EsMV becomes smaller near resonance.

Response: We thank the reviewer for this thoughtful and precise question. As correctly noted, when the input current becomes sufficiently large, the relative influence of the coupled permanent magnets diminishes and the electromagnetic force becomes dominant. In this high-current regime, the maximum vibration amplitude of the NC-EsMV can indeed approach that of the C-EsMV, consistent with the convergence observed near resonance in Fig. 1f.

The key distinction, however, lies in the operating regime and energy-conversion efficiency. The C-EsMV achieves large-amplitude vibration at significantly lower input currents, whereas the NC-EsMV remains confined to small amplitudes until very high currents are applied. Beyond this point, the NC-EsMV experiences pronounced Joule heating and a rapid drop in conversion efficiency (Supplementary Fig. 8).

Here, energy-conversion efficiency refers to the fraction of electrical input energy that is effectively transformed into the kinetic and elastic energy of vibration. This definition highlights that the coupled configuration enables more efficient utilization of input power through cyclic storage and release via the elastic–magnetic interaction.

Furthermore, the NC-EsMV primarily relies on linear resonance to achieve large amplitudes, whereas the C-EsMV exploits elasto-magnetic instability to realize nonlinear amplification even in the absence of resonance. This instability-driven response accounts for the mechanical-memory behavior discussed later in the manuscript and cannot be replicated simply by increasing the current or tuning the excitation frequency in the NC configuration.

Comment: 4. The manuscript repeatedly discusses energy efficiency, but there is no clear definition or description of how it is calculated. This should be explicitly provided in the main text.

Response: We thank the reviewer for pointing out the need to clarify the definition of “energy efficiency.”

As noted in our responses above, the term refers specifically to energy-conversion efficiency—the proportion of

electrical energy supplied to the electromagnet that is converted into the kinetic and elastic energy of vibration.

In the NC-EsMV, only a small fraction of the input energy contributes to motion due to weak magnetic coupling, whereas in the C-EsMV, magnetic interaction allows a greater portion of the input to be stored as elastic energy and released as kinetic energy during collapse–release cycles. This results in a conversion efficiency over 700-fold higher than that of NC-EsMV (Supplementary Note).

In the revised manuscript, we have replaced “energy efficiency” with “conversion efficiency,” explicitly defined it in the main text, and added a brief description of the calculation method for clarity.

Modification/Addition

1. Modification of “energy efficiency” to clarify it as “energy-conversion efficiency” and to provide its definition.

1) In page 2, line 53.

~ capable of amplifying kinetic energy by over three orders of magnitude compared to a non-coupled control system (NC-EsMV) (Fig. 1a and b).

→ ~ capable of **enhancing kinetic energy conversion** by over three orders of magnitude compared to a non-coupled control system (NC-EsMV) (Fig. 1a and b).

2) In page 6, line 152. (original content moved to Supplementary Fig. 5)

This stands in sharp contrast to the C-EsMV system, which undergoes four distinct stages in each vibrational cycle: i) loading, ii) take-off, iii) vibration, and iv) landing, marked by vertical dashed lines. Initially, the magnet collapses onto the electromagnet’s surface, storing maximum elastic potential energy (i, $0 \text{ s} < t < 0.077 \text{ s}$). When electromagnetic repulsion increases sufficiently to yield a net positive force, it triggers the take-off (ii, $t = 0.077 \text{ s}$). After take-off, the system enters a vibration phase governed by its natural frequency (iii, $0.102 \text{ s} < t < 0.132 \text{ s}$), then returns to the electromagnet’s surface to complete the cycle (iv, $0.132 \text{ s} \leq t \leq 0.2 \text{ s}$). In the C-EsMV system, magnetic attraction establishes a critical threshold that facilitates significant elastic to kinetic energy exchange.

→ **In contrast, the C-EsMV system establishes a critical threshold that enables efficient conversion of electrical input into mechanical motion. This configuration enables improved conversion efficiency through elasto–magnetic coupling, allowing a larger portion of the supplied power to be stored and released as kinetic energy.**

3) In page 9, line 210.

we analyzed the ratio of the maximum kinetic energy of C-EsMV to NC-EsMV ($\eta_{C/NC}$) as a function of I_{peak} with $D = 4.0$ mm.

→ we analyzed the **ratio of energy conversion efficiencies ($\epsilon_{C/NC}$, the ratio of the maximum kinetic energy of C-EsMV to that of NC-EsMV under identical electrical input)** as a function of I_{peak} with $D = 4.0$ mm.

4) In page 10, line 257.

System efficiency can be enhanced by waveform design (Supplementary Note 5, Fig. 3i), as the waveform shape directly affects the input energy required for amplified vibrations.

→ **The energy conversion efficiency of the system—defined as the ratio of the kinetic energy of the vibrating magnet to the supplied electrical energy—is relatively low (< 1 %) because of the limited coil turns and associated geometric factors of the electromagnet. However, this efficiency can be substantially enhanced by waveform design (Supplementary Note 6, Fig. 3i), as the waveform shape directly affects the input energy required for amplified vibrations.**

Comment: 5. Regarding Figure 4f (right side, asymmetric vibration), were D_R , D_L , and H_L all changed simultaneously? Were only three separate groups tested? Please clarify.

Response: We thank the reviewer for this detailed question. The experiment in Fig. 4f aimed to demonstrate that the C-EsMV system also operate under asymmetric magnetic coupling, where asymmetry allows the actuation force to be concentrated on one side.

In this study, three distinct systems were tested:

- **System 1 (asymmetric, $D_L > D_R$):** (D_L, D_R, h_L, h_R) = (4.0, 3.0, 1.5, 1.5)
- **System 2 (symmetric):** (D_L, D_R, h_L, h_R) = (3.5, 3.5, 1.5, 1.5)
- **System 3 (asymmetric, $D_L < D_R$):** (D_L, D_R, h_L, h_R) = (3.0, 5.0, 2.5, 1.5)

In System 3, the right-side spacer thickness (D_R) was increased so that, upon detachment, the magnet–membrane pair could store more elastic energy and release it as a stronger rebound force—analogueous to a slingshot effect. However, increasing D_R also weakens the magnetic attraction on that side. To compensate, the left-side magnet was

made thicker ($h_L = 2.5$ mm, $h_R = 1.5$ mm) and positioned closer to the electromagnet ($D_L < D_R$) to strengthen its magnetic contribution.

Thus, in each configuration both the spacer thickness (D) and magnet thickness (h) on each side were adjusted simultaneously, and the data represent three distinct experimental sets. This clarification has been added to the revised manuscript for accuracy and clarity.

Modification/Addition

1. Addition to the manuscript clarifying the experimental design of the asymmetric impact force test.

- In page 12, line 292.

To maintain overall balance, the opposite side was adjusted by increasing its magnet thickness and reducing its gap, compensating for the weakened magnetic attraction. Each configuration was tested as an independent system, confirming that asymmetric coupling can effectively localize and enhance actuation on a chosen side.

Comment: 6. What exactly is meant by the membrane's natural frequency? Will this change when the modulus or other parameters of the membrane are varied?

Response: We thank the reviewer for raising this important point. In the original manuscript, the term natural frequency was used to describe the resonance at which the system exhibits maximum vibration amplitude under AC excitation. Strictly speaking, this is not the intrinsic free-vibration frequency of the membrane–magnet pair but the nonlinear forced resonance of the coupled elasto-magnetic system. We have clarified this distinction in the revised text.

The intrinsic baseline frequency of the magnet–membrane pair in the absence of magnetic and damping forces can be given by

$$\omega_0 = \sqrt{\frac{k_{ES}}{m}},$$

where k_{ES} is the elastic membrane spring constant and m is the magnet–membrane mass. When magnetic and damping forces are included, the damped frequency becomes

$$\omega_d = \sqrt{\frac{k_{\text{ef}}(z^*)}{m} - \left(\frac{c_{\text{ef}}}{2m}\right)^2},$$

where c_{ef} denotes the damping coefficient of the membrane, and $k(z^*)$ varies with the equilibrium position z^* .

Therefore, the observed resonance in our experiments corresponds to a nonlinear forced response whose frequency shifts dynamically with the position and the applied current, rather than a single fixed natural frequency.

While the membrane can support higher-order vibration modes, polymer damping in our experimental regime suppresses these modes, leaving the response dominated by the fundamental resonance. We have revised the main text to explicitly define this distinction and included the above relations for clarity.

Modification/Addition

1. Addition of the explanation of resonance frequency of our system.

- In page 9 of Supplementary materials.

Supplementary Note 3. Resonant frequency in our system

In our system, the observed resonance corresponds to the nonlinear forced resonance of the coupled magnet–membrane system rather than the intrinsic free-vibration frequency. Consequently, the response around resonance is asymmetric: the vibration amplitude increases gradually as the frequency approaches resonance, but exhibits an eventual decay beyond the resonance frequency. At lower frequencies, the system surpasses the threshold to generate amplified vibration, and as the frequency approaches resonance, this effect further enhances the amplitude. Beyond resonance, however, the increasing phase mismatch between the magnet motion and the input signal reduces effective acceleration, preventing the system from overcoming the threshold and leading to weakened vibration and eventual decay.

This asymmetry primarily arises from the phase mismatch–induced loss of energy transfer efficiency at higher frequencies. In addition, the strongly nonlinear magnetic force, which enhances as the inter-magnet distance decreases, further intensifies this asymmetric response. Together, these effects cause the frequency response to deviate from the symmetric Lorentz-type profile typical of linear oscillators.

The intrinsic baseline frequency of the magnet–membrane pair, in the absence of magnetic and damping forces, can be given by the mass–spring relation:

$$\omega_0 = \sqrt{\frac{k_{\text{ES}}}{m}} \quad (9)$$

where k_{ES} is the spring constant of membrane and m is the magnet–membrane mass.

When magnetic and damping forces are included, the system can be linearized locally around an equilibrium position z^* :

$$m z'' + c_{\text{ef}} z' + k_{\text{ef}} z = 0, \quad (10)$$

$$k_{\text{ef}} = \frac{\delta(F_{ES} - F_{MM})}{\delta z} \Big|_{z=z^*},$$

The corresponding damped frequency is given by:

$$\omega_d = \sqrt{\frac{k_{\text{ef}}(z^*)}{m} - \left(\frac{c_{\text{ef}}}{2m}\right)^2}, \quad (11)$$

where c_{eff} denotes the effective damping coefficient of the membrane, which in our system may depend not only on velocity but also on displacement due to the viscoelastic membrane behavior and contact conditions. $k_{\text{eff}}(z^*)$ varies dynamically with the equilibrium position determined by the applied magnetic forces. Thus, the resonant frequency shifts continuously with both equilibrium position and input current, confirming that the observed resonance arises from a nonlinear forced response rather than a fixed natural frequency.

For small oscillations around an equilibrium z^* , $c(z)$ can be approximated as constant, yielding a well-defined local damped natural frequency. However, in the amplified state, the C-EsMV traverses both equilibria with large excursions, where the restoring and damping forces vary significantly within each cycle. The motion is therefore better described as a nonlinear forced periodic response (limit cycle) governed by the AC electromagnetic drive, whose magnitude depends on displacement while its temporal variation follows the input frequency. This displacement dependence introduces nonlinear stiffness effects, broadening and shifting the resonance relative to eq. (9).

Although the membrane can, in principle, support multiple vibration modes, our experiments operated near the fundamental resonance, where phase mismatch and potential damping effect of membrane strongly suppresses higher-order responses. Weak secondary resonances may appear under certain conditions but are highly sensitive to asymmetry and nonlinear coupling. If geometric or material asymmetry were introduced, multimode or non-axisymmetric oscillations could emerge, leading to richer nonlinear dynamics under modified designs or excitation conditions.

Comment: 7. Figure 5, which presents four memory regions, is highly interesting. However, the figure contains too much information while the mechanistic discussion is too limited, making it difficult to clearly understand the results.

Response: We are grateful to the reviewer for this valuable suggestion regarding Fig. 5. We agree that the original version contained excessive detail, which made interpretation less straightforward. In the revised manuscript, we have simplified the figure by removing secondary information and retaining only the essential features.

In addition, we have expanded the mechanistic discussion in the main text to clarify how the four memory regimes arise and how they relate to the observed hysteresis behavior. We believe these changes improve both the clarity and readability of the results.

Modification/Addition

1. Addition of a Supplementary Figure and related context describing the shooting behavior and dynamic memorization.

- In page 28 of Supplementary materials.

Supplementary Fig. 13. Numerical results of electrical shooting behavior and motion hysteresis in C-EsMV.

(a) Vibration motion of NC-EsMV and C-EsMV at a driving frequency of 5 Hz ($D = 4.0$ mm). Even after the current is increased from 1.4 A to 2.0 A and then returned to 1.4 A, the amplified vibration of C-EsMV is maintained. The shooting process enables C-EsMV to reach the amplified mode at lower currents that were previously unattainable, as clearly seen in the phase portraits. (b–d) Static and dynamic vibration paths at $I_{\text{peak}} = 1.4$ A before shooting: (b) Bifurcation diagram of C-EsMV showing equilibrium positions, (c) corresponding phase portraits of C-EsMV and NC-EsMV, and (d) dynamic vibration trajectory under sinusoidal electrical input, which closely follows the equilibrium path.

(e–g) Static and dynamic vibration paths at $I_{\text{peak}} = 1.4$ A after shooting: (e) Bifurcation diagram of C-EsMV showing a switched branch after shooting, (f) phase portraits of C-EsMV and NC-EsMV, and (g) dynamic vibration trajectory under repeated sinusoidal input, exhibiting significantly larger oscillations despite the same current amplitude.

- In page 9, line 221.

- This triggering process, which we term “shooting,” refers to a momentary stimulus that propels the magnet across the energy barrier separating the coexisting states (*weakened and amplified*). As shown in Supplementary Fig. 12, when the

input current was increased above the onset threshold ($I_{th,on}$) and then reduced (shooting), the system exhibited a sudden transition from a weakened to an amplified vibration state, maintaining the large amplitude even after the current returned to the initial level ($I_{peak} \approx 1.3$ A). In contrast, systems that never exceeded $I_{th,on}$ (pre-shooting) remained confined to low-amplitude oscillations because the available energy was insufficient to overcome the potential barrier (i.e., the basin boundary between the two attractors). Experimentally, a brief electrical trigger induced this dynamic shift, confirming that the vibration state could be switched and retained through a transient perturbation.

This behavior was further validated through simulation (Supplementary Fig. 13). In the model, a short high-current pulse ($I_{peak} = 2.0$ A) served as the electrical trigger, driving the system across the potential barrier. Comparing the bifurcation diagrams before and after shooting at $I_{peak} = 1.4$ A revealed that the response followed a distinct path along the upper stable branch, resulting in a larger limit-cycle oscillation even in the absence of resonance. The persistence of this amplified vibration after returning to the lower input current arises from inertia, which provides residual kinetic energy to sustain oscillations around the new equilibrium branch. This inertia-driven retention is stably maintained and was further validated as an efficient energy conversion mechanism (Supplementary Fig. 14). It constitutes the vibrational hysteresis-based mechanical memory, which is further analyzed in Fig. 5.

- In page 37 of Supplementary materials.

Supplementary Fig. 22. Numerical verification of mechanically triggered amplification.

Vibration motion of NC-EsMV and C-EsMV systems at an electromagnet-to-magnet distance of (a) $D = 4.0$ mm and (b) $D = 4.5$ mm, both under an input current of 2.0 A. (c) Analogous to the electrical shooting observed in Supplementary Fig. 13, mechanical triggering ($D = 4.5$ mm \rightarrow 4.0 mm \rightarrow 4.5 mm) also sustains amplified vibration in C-EsMV after the trigger is removed, with constant input current. (d) Phase portraits of each system. The C-EsMV exhibits a larger-amplitude limit cycle (bold line) at the same input current and D , indicating greater vibration velocity and displacement compared with NC-EsMV.

- In page 38 of Supplementary materials.

Supplementary Fig. 23. Mechanism of vibrational hysteresis behavior in mechanical memory.

Under identical input current I_{peak} , inertia allows two distinct vibration responses: (a) Pre-shooting (weakened state) and (b) post-shooting (amplified state). Without shooting, the magnet oscillates with small displacement near z_{off} (positions 1 – 3) unable to overcome the potential barrier imposed by magnetic attraction. As a result, the system remains in a low-amplitude vibration regime (Point A). By contrast, a brief perturbation (shooting)—which can be electrical, magnetic, or mechanical—pushes the magnet across the barrier to the bottom surface ($z = 0$, position “2”), storing additional elastic energy in the membrane. During the subsequent repulsive phase, this stored energy is released as a large overshoot (“3 \rightarrow 4”), and the magnet returns with sufficient inertia to repeatedly cross the barrier (“4 \rightarrow 1 \rightarrow 5 (= 2)”). The residual kinetic energy from each cycle sustains these large-amplitude oscillations even after the external trigger is removed, corresponding to Point B on the amplitude– I_{peak} curve. If this residual energy is intentionally dissipated (see Supplementary Video 7), the system reverts to the weakened state. Potential energy–position graphs are based on Supplementary Fig. 9e.

- In page 12, line 309.
 - This process is analogous to the electrical shooting described earlier, relying on inertia–elastic energy exchange that drives the transition from the weakened to the amplified vibration state (Supplementary Fig. 23). Once triggered, the system sustains oscillation within the amplified branch even after the external input is removed. As shown in Supplementary Fig. 24, this design effectively stores external mechanical stimuli as persistent amplified vibrational states—a phenomenon we term mechanical memory.
- In page 42 of Supplementary materials.

Supplementary Fig. 27. Effect of membrane damping on mechanical memory.

(a) $\eta = 50$ (b) $\eta = 500$ (c) $\eta = 5000$ (d) $\eta = 50000$ Pa·s.

- In page 13, line 332.
 - However, the duration of this non-volatile retention is governed by the viscoelastic property of the membrane (Supplementary Fig. 27): as discussed earlier, higher viscosity increases damping and gradually suppresses sustained oscillations, whereas a more elastic membrane maintains stable amplified vibration. This confirms that membrane elasticity is essential for robust memory retention.

2. Modification of Fig. 5 for improved clarity and readability.

1) Original version

2) Revised version (To reduce visual density, we removed the excessive explanatory text.)

Comment: 8. Some figures and captions are unclear, and the authors are encouraged to review and improve them.

for example:

a) in Figure 1c and d, what do the inserted physical photos represent?

Response: We thank the reviewer for this detailed comment regarding the figure. The inserted photos in Fig. 1c and 1d are intended to show the actual positions of the magnets corresponding to the equilibrium points indicated by the green and red dots in the potential energy landscape. To improve clarity, we have revised the figure and caption to make this correspondence more explicit.

Modification/Addition

1. Modification of Fig 1c and d for explanation of physical photos.

1) Original version

2) Revised version

Comment: 8.b) In Figure 2, the font size is too small and the grey color makes it hard to read. The many overlapping lines obscure the key point (e.g., inertial under/overshoot).

Response: We sincerely appreciate the reviewer’s careful observation of this figure. To improve readability, we have increased the font size, adjusted the color scheme to enhance contrast, and reduced overlapping lines in Fig. 2 so that the key features (including inertial under/overshoot) are clearer.

Modification/Addition

1. Modification of Fig 2 for clarity and readability.

1) Original version

2) Revised version (To reduce visual density, we removed the inset graph and arrows to simplify the figure)

Comment: 8. c) In Figure 3i, the vertical axis “ η/η_{sine} ” needs definition.

Response: We appreciate the reviewer’s careful attention to detail. The vertical axis label “ η/η_{sine} ” has now been clearly defined in the main text, and the corresponding explanation has been added to ensure clarity. To avoid confusion caused by the repeated use of the symbol η for the dynamic viscosity, we have replaced it with “ ϵ ” throughout the manuscript.

Modification/Addition

1. Modification of Fig. 3i for clarification of the energy conversion efficiency compared to a sine wave, ϵ/ϵ_{sine} .

1) In page 11, line 264.

Notably, concave waveforms, particularly pseudo-gaussian type, offer significantly higher energy efficiency because of reduced energy consumption. Simulations show that a pseudo-gaussian waveform can improve efficiency by roughly 64.4 times compared to a sine wave, ~

→ Notably, concave waveforms, particularly pseudo-gaussian type, offer significantly higher energy **conversion**

efficiency because of reduced energy consumption. Simulations show that a pseudo-gaussian waveform can improve efficiency by roughly 64.4 times compared to a sine wave (ϵ/ϵ_{sine}).

Comment: 9. Numerical formatting should be consistent: e.g., line 306 states “70,000% increase in energy efficiency,” whereas line 195 states “700”; please unify style.

Response: We appreciate the reviewer’s valuable observation regarding numerical consistency. We have revised the manuscript to ensure a uniform style, and all values are now consistently reported using the “700-fold” format.

Modification/Addition

1. Modification of expressions to unify manuscript style.

- In page 14, line 356.

, leads to vigorous vibrations and efficient energy release, achieving up to a 70,000% increase in energy efficiency relative to the NC-EsMV system.

→ , leads to vigorous vibrations and efficient energy release, achieving up to a **700-fold** increase in energy efficiency relative to the NC-EsMV system.

Comment: 10. It is suggested that recent related work be cited, for example:

Y. Tian, Adv. Funct. Mater. (2025), “A Dynamically Programmable Hydrogel Surface with Rapid Magnetically Actuated Snapping of Bistable Dome Configurations” (doi:10.1002/adfm.202508885).

Response: We are grateful to the reviewer for pointing out this excellent and highly relevant study. We have added the suggested reference to our revised manuscript.

Modification/Addition

1. Addition of the following references.

- In page 18 (References).

18. Tian, Y. et al. A dynamically programmable hydrogel surface with rapid magnetically actuated snapping of bistable dome configurations. *Adv. Funct. Mater.* **35**, 2508885 (2025).